# A novel framework for educational Q&A: Leveraging RAG and Code Interpreters for knowledge retrieval and logical computation

Jin Lu[1], Ji Li[2]*

**1** Guangdong Key Laboratory of Big Data Intelligence for Vocational Education, Shenzhen Polytechnic University, Shenzhen, Guangdong, China, **2** Research Management Office, Shenzhen Polytechnic University, Shenzhen, Guangdong, China

\* liji@szpu.edu.cn

## Abstract

This paper presents a novel approach to enhancing educational question-answering (Q&A) systems by combining Retrieval-Augmented Generation (RAG) with Large Language Model (LLM) Code Interpreters. Traditional educational Q&A systems face challenges in areas such as knowledge updates, reasoning accuracy, and the handling of complex computational tasks. These limitations are particularly evident in domains requiring multi-step reasoning or access to real-time, domain-specific knowledge. To address these issues, we propose a system that utilizes RAG to dynamically retrieve up-to-date, relevant information from external knowledge sources, thus mitigating the common "hallucination" problem in LLMs. Additionally, the integration of an LLM Code Interpreter enables the system to perform multi-step logical reasoning and execute Python code for precise calculations, significantly improving its ability to solve mathematical problems and handle complex queries. We evaluated our proposed system on five educational datasets—AI2_ARC, OpenBookQA, E-EVAL, TQA, and ScienceQA—which represent diverse question types and domains. Compared to vanilla Large Language Models (LLMs), our approach combining Retrieval-Augmented Generation (RAG) with Code Interpreters achieved an average accuracy improvement of 10−15 percentage points. Among tested models, GPT-4o and Gemini-pro-1.5 consistently showed the strongest performance, excelling particularly in scientific reasoning, multi-step computations. Despite these advancements, we identify several challenges that remain, including knowledge retrieval failures, code execution errors, difficulties in synthesizing cross-disciplinary information, and limitations in multi-modal reasoning, particularly when combining text and images. These challenges provide important directions for future research aimed at further optimizing educational Q&A systems. Our work shows that integrating RAG and Code Interpreters offers a promising path toward more accurate, transparent, and personalized

**Data availability statement:** All relevant data are within the paper and its Supporting information files.

**Funding:** 2025 Guangdong Philosophy and Social Science Planning Project "Research on the Synergistic Evolutionary Mechanisms of the Governance System of Colleges and Universities and the Development of Students' Socio-emotional Competence: Based on Multimodal Learning Analysis, GD25CJY29.

**Competing interests:** The authors have declared that no competing interests exist.

educational Q&A systems, and can significantly improve the learning experience in various educational contexts.

## 1. Introduction

The educational question-answering system is playing an increasingly important role in modern education [1]. With the increasing demand for personalized learning, students not only expect tailored guidance according to their individual learning progress and comprehension levels, but also require timely feedback and assistance during the learning process, especially when there is insufficient teacher support. Traditional teaching models, which rely on face-to-face instruction, often fail to meet the immediate needs of large student populations, particularly during peak periods or in resource-limited educational environments. Therefore, automated educational question-answering systems have become crucial tools to fill this gap, aiming to enhance students' learning efficiency and outcomes by providing accurate and personalized responses [2]. However, traditional machine learning-based educational question-answering systems still face significant challenges in terms of knowledge updates and reasoning capabilities [3]. First, traditional machine learning models typically rely on static datasets for training, which have limitations in terms of timeliness and comprehensiveness of knowledge, making them inadequate for rapidly evolving domains. For example, advancements in science and technology, emerging social events, and the continuous updating of academic knowledge all demand that educational question-answering systems dynamically acquire and integrate the latest information. Second, traditional models have limited reasoning abilities and struggle to handle complex logic or mathematical reasoning tasks. This limitation is especially evident when facing problems that require multi-step calculations, rigorous logical deductions, or deep comprehension. Furthermore, traditional systems perform poorly when handling open-ended questions, often failing to provide flexible and effective solutions, which compromises the accuracy and relevance of the answers.

In the past two years, the rapid development of large language models (LLMs) such as GPT-4, Llama-3, and Gemma-2 has brought unprecedented possibilities for educational question-answering systems [4–7]. These models, based on deep learning and large-scale pre-training, possess powerful natural language processing capabilities, enabling them to generate fluent and highly contextually relevant responses, significantly improving the quality of the system's answers. Especially in personalized learning, LLMs can adjust their answers according to the student's actual needs and background, helping them better grasp knowledge. For example, when a student is learning mathematics, the LLM can provide explanations and practice problems at varying levels of difficulty according to the student's current understanding, promoting gradual improvement. However, despite their excellence in text generation and comprehension, LLMs still face limitations in handling complex reasoning and mathematical calculations, and they are susceptible to the "hallucination" problem, where the model may generate incorrect answers that do not align with actual knowledge, posing challenges to the accuracy and reliability of the educational system [8,9].

To address these challenges in current educational question-answering systems, this study proposes a new LLM-based educational question-answering system that combines Retrieval-Augmented Generation (RAG) and LLM Code Interpreter [10–16]. RAG is an innovative framework that integrates information retrieval [17] (IR) with generative models (such as Transformer architectures) to assist in generating more accurate and relevant answers by retrieving information from external knowledge bases. Specifically, RAG first retrieves relevant external textual resources and then generates answers based on the retrieved information, effectively reducing the hallucination problem of models [16]. Traditional large language models, when lacking external data support, may generate false information, while RAG dynamically acquires relevant information from external knowledge bases, greatly enhancing the accuracy and timeliness of the answers and ensuring that the system can handle a wide range of knowledge domains and emerging issues. The working principle of RAG involves two main steps: first, the system uses a retrieval module to extract relevant text segments from large text corpora or domain-specific knowledge bases that are related to the user's question. These text segments typically cover the core information and background knowledge of the question, providing a solid foundation for the generative module. Second, the generative module uses the retrieved text segments as input and generates highly relevant and accurate answers using pre-trained generative models (e.g., GPT-4). This dual mechanism not only performs well in answering simple questions but also handles more complex open-ended questions, improving the system's overall performance and adaptability. For example, when a student asks a question related to the latest scientific research, RAG can retrieve the latest research results and generate an accurate and authoritative response based on them.

In addition, the introduction of the LLM Code Interpreter adds reasoning and calculation capabilities to the system [15,16]. While traditional large language models excel at generating natural language text, they fall short when it comes to handling mathematical problems, logical reasoning, and executing code. The LLM Code Interpreter is capable of explaining and executing generated code snippets, providing the educational question-answering system with the ability to perform calculations and logical reasoning. This enhancement allows the system not only to answer simple question-based inquiries but also to handle tasks involving complex calculations and reasoning. For example, when a student asks a math problem requiring multi-step calculations, the LLM Code Interpreter can dynamically generate and execute the corresponding mathematical operations, providing accurate solutions. Similarly, for logical reasoning problems, the system can validate the reasoning process by explaining and running code, offering more rigorous and reliable answers.

By combining the advantages of RAG and the LLM Code Interpreter, the educational question-answering system proposed in this study enhances user experience and answer quality on multiple levels. First, RAG effectively addresses the hallucination problem of large language models [18,19], ensuring the accuracy and reliability of the system's answers. Second, the introduction of the LLM Code Interpreter strengthens the system's capabilities in calculations and reasoning, especially in handling mathematical and complex logical tasks. This innovation enables the educational question-answering system to be more adaptable when faced with a diverse range of questions, providing learners with more personalized, dynamic, and authoritative responses. For example, when students are studying calculus, the system can not only provide theoretical explanations but also use the Code Interpreter to generate specific calculation steps, helping students understand and master complex mathematical concepts and methods.

In our educational question-answering system, as shown in Fig 1, the RAG (Retrieval-Augmented Generation) module and LLM Code Interpreter work together to generate precise and efficient answers. When a user submits a question, the system first prepossess the input, including text normalization and formatting. These prepossessing steps ensure that the question is correctly parsed, facilitating subsequent knowledge retrieval and processing by the generation model. At this stage, the question may involve multi-step reasoning, external knowledge, or require computational tasks. Next, the RAG module is activated, responsible for retrieving relevant literature, document fragments, or knowledge content from external knowledge bases. The retrieval mechanism generates a query vector from the user's question and performs similarity calculations with the texts in the knowledge base to find the most relevant documents. To enhance the timeliness and accuracy of the knowledge, the knowledge base includes a wide range of content from encyclopedias, textbooks, scientific

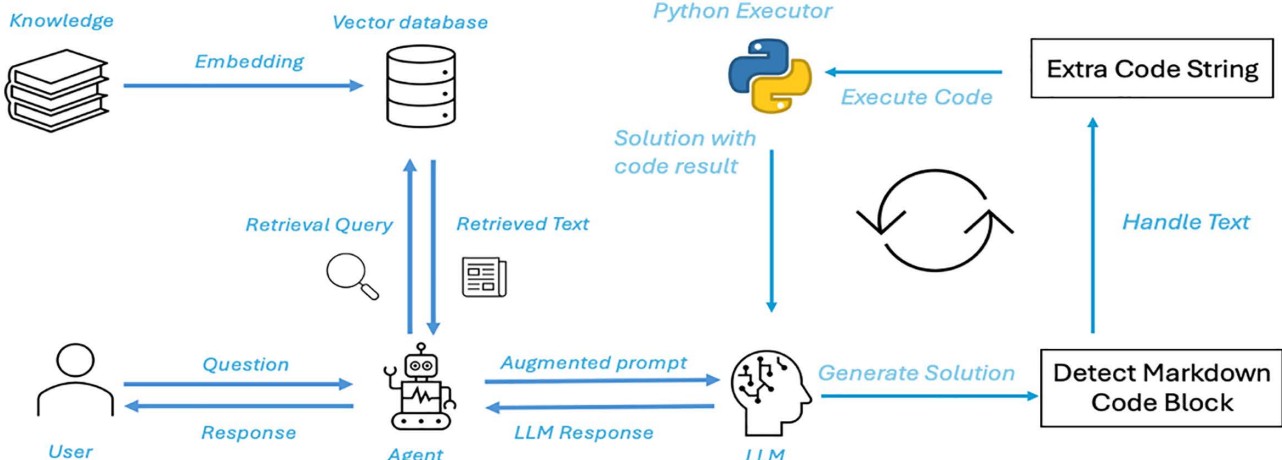

**Fig 1. Overview of the proposed educational question-answering (Q&A) system integrating Retrieval-Augmented Generation (RAG) and Large Language Model (LLM) Code Interpreter.**

papers, and more. The key goal of this process is to ensure that the generated answer is built on the most up-to-date and relevant knowledge, avoiding the hallucination issues common in traditional LLMs. The retrieved information, along with the original user query, is then input into the LLM. At this stage, the LLM generates the answer based on the user's question and the retrieved external knowledge. The system utilizes the language model's powerful comprehension and reasoning abilities, combining this with the external information to produce the response. If the question involves multi-step reasoning or requires complex computations, the generated answer may trigger the use of the code interpreter. When the LLM's generated answer requires the execution of a program, the system automatically activates the LLM Code Interpreter. In this case, the LLM generates the corresponding code (such as Python code) to execute the reasoning process. The code interpreter runs this code within a sandbox environment, ensuring the safety of the computational and reasoning steps. Once the code execution is successful, the calculation results from the interpreter and the LLM-generated text answer are integrated (this process can repeat multiple times, executing several blocks of code), forming the final system response. The system then delivers the final answer to the user, while also displaying the relevant reasoning process or code execution logs to increase the answer's venerability and understanding. For multi-step calculation problems, the system shows intermediate results at each step to help the user follow the problem-solving process. Furthermore, if the user has questions about any part of the answer, the system supports follow-up queries, allowing the user to retrieve additional information or run more reasoning steps. RAG and the code interpreter, the educational question-answering system is able to provide answers that are both rich in knowledge and up-to-date, while also handling complex computational and reasoning tasks. RAG ensures the system answers based on the most current knowledge, while the code interpreter allows for the precise execution of multi-step reasoning and computational tasks.

The findings of this study provide new insights and methodologies for enhancing educational question-answering systems, demonstrating substantial improvements over vanilla Large Language Models (LLMs). Specifically, by integrating Retrieval-Augmented Generation (RAG) and the LLM Code Interpreter, our system achieves an average accuracy increase of 10–15 percentage points compared to vanilla LLM. This approach effectively addresses key limitations of vanilla LLMs, such as knowledge hallucination, limited computational capabilities, and inadequate reasoning performance. As technology continues to advance, this integrated framework is expected to play an increasingly critical role in personalized education and intelligent tutoring, further promoting educational equity and significantly improving educational quality.

## 2 Related works

### 2.1 Knowledge Base Question Answering Systems (KBQA)

Past educational question answering systems primarily relied on Knowledge Base Question Answering Systems [20–23] (KBQA), which respond to user queries using per-constructed, structured knowledge bases (such as Freebase [24], Wikidata [25], etc.). These systems are typically built upon Information Retrieval [17] (IR) techniques or Knowledge Graphs [26] (KG). The system first parses the user's natural language question, identifies the entities, relationships, and attributes within it, and then queries the constructed knowledge graph based on this information. In this manner, the system can extract relevant information from the predefined knowledge base and generate answers by incorporating the context of the query. In specific domains (such as medicine, law, corporate knowledge, etc.), KBQA systems are generally able to provide relatively accurate answers, particularly when dealing with structured and standardized questions, showcasing high accuracy in responses [27].

However, traditional KBQA systems exhibit significant limitations. First, the construction and maintenance of the knowledge base often rely on substantial manual labor and rule-making. Especially in dynamic fields, the update cycle of the knowledge base typically cannot keep pace with real-world changes [28]. For example, when new academic research or industry developments emerge, the relevant knowledge bases may fail to reflect these changes in time, causing the system to be unable to provide the most up-to-date information [21]. Second, methods based on structured knowledge bases still face considerable challenges in dealing with the diversity and complexity of natural language [22]. Natural language contains a large number of ambiguous, uncertain, or polysemous words, and in open-domain questions, traditional knowledge bases often struggle to handle this diversity effectively. When confronted with complex contexts or phenomena such as polysemy and metaphor, traditional KBQA systems tend to show insufficient accuracy and relevance in their answers. Moreover, traditional KBQA systems lack flexibility and adaptability, especially when addressing cross-disciplinary, cross-domain, or novel questions. [23] In such cases, the system's performance is often unsatisfactory, unable to provide comprehensive and timely answers. In conclusion, these issues limit the effective application of traditional knowledge base question answering systems in broad educational contexts, particularly when handling complex, variable natural language queries.

### 2.2 Retrieval-Augmented Generation (RAG)

With the rise of large language models [4] (LLMs), particularly driven by models such as GPT-4, Llama-3, and Gemma-2, new opportunities have emerged for educational question answering systems. LLMs, through deep learning techniques, are capable of understanding and generating natural language, significantly enhancing the fluency and context relevance of question answering systems. The advantages of LLMs in text generation allow them to generate high-quality answers, enabling them to respond flexibly to various natural language queries. However, despite LLMs' outstanding performance in language understanding and generation, they still face challenges when dealing with tasks that require complex reasoning, rigorous logic, or precise calculations. LLMs primarily rely on pattern matching and probabilistic generation, lacking true logical reasoning capabilities, making them ill-equipped to handle problems that involve multi-step reasoning, strict calculations, or problem-solving. In such cases, LLMs may generate answers that do not align with actual knowledge, or they may produce "hallucinations" i.e., incorrect or fabricated information.

To address the limitations of LLMs in reasoning and accuracy, the Retrieval-Augmented Generation [29] (RAG) method has been introduced, as shown in Fig 2. RAG combines the strengths of information retrieval and generative models. Before generating an answer, it first retrieves relevant information from external unstructured textual data related to the user's query and uses this information as context input for the generative model, thereby improving the quality of the generated answer. Specifically, RAG combines a retrieval module with a generation module: it first performs information retrieval to identify the most relevant documents or knowledge fragments and then passes this retrieved information to

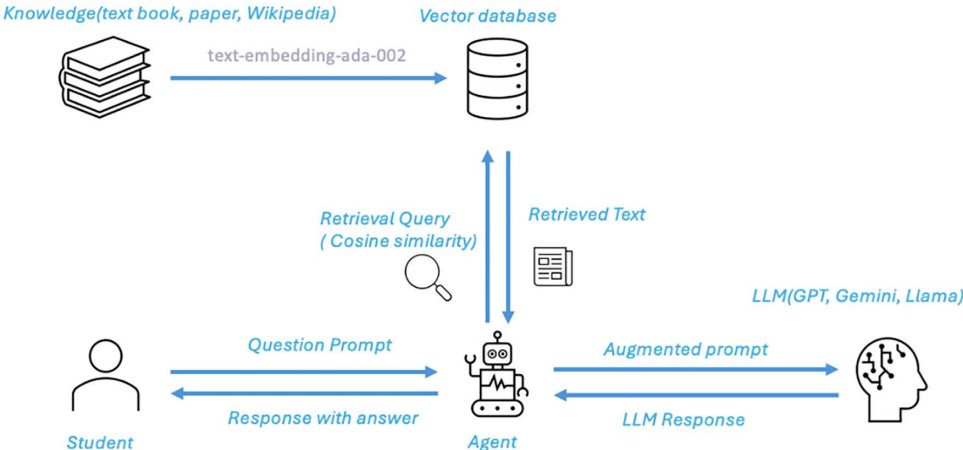

**Fig 2. How RAG work.**

the LLM to help it generate more precise and informative answers. This mechanism of RAG effectively reduces the risk of LLMs generating false information, especially in open-domain question answering, where it can dynamically leverage the latest information from external knowledge bases, enhancing the system's adaptability to dynamic knowledge domains.

Although RAG has made progress in alleviating the "hallucination" problem of LLMs, it still faces certain limitations when handling complex mathematical computations and logical reasoning tasks [30]. Since the generative model itself does not possess strong mathematical computation capabilities [31], it may still generate inaccurate or erroneous results when confronted with tasks that require multi-step calculations or precise answers [32]. For example, when processing mathematical problems, although RAG can provide related formulas and methods, it cannot perform actual computations, potentially leading to inaccuracies when dealing with complex reasoning or precise numerical calculations. While RAG can mitigate this issue by leveraging external information to some extent, systems that rely on external textual information still face challenges in terms of precision and reliability in certain tasks.

### 2.3 Large Language Models (LLMs) and Code Interpreters

To address the limitations of LLMs in reasoning and computation, researchers have gradually introduced the Code Interpreter component [16,33]. A Code Interpreter is a tool that can generate and execute code, allowing it to handle complex tasks such as mathematical operations [34], data manipulation [35], and logical reasoning [36]. Unlike traditional text-based generation methods, a Code Interpreter directly processes computation tasks using programming languages, thereby overcoming the shortcomings of LLMs in complex computational tasks, as shown in Fig 3. By generating code and executing it in real-time, the LLM can obtain precise calculation results and detailed reasoning processes. For example, when a student presents a complex mathematical problem, the system can generate code snippets (such as Python code) and execute them in a controlled environment to obtain accurate answers and provide them to the user. In this way, the Code Interpreter not only improves the system's performance in handling complex tasks but also enhances the transparency and venerability of the answers. Users can view the code executed by the system and verify each step of the calculation process, thereby increasing the system's credibility and interoperability.

Additionally, the Code Interpreter plays an important role in logical reasoning tasks. Many problems involving logical reasoning (such as mathematical proofs, algorithm design, etc.) require multi-step reasoning to reach a conclusion. Traditional LLMs often rely on guessing and probabilistic generation when handling such tasks, which can lead to errors in the reasoning process. The Code Interpreter can directly execute logical code to verify each step of the reasoning process,

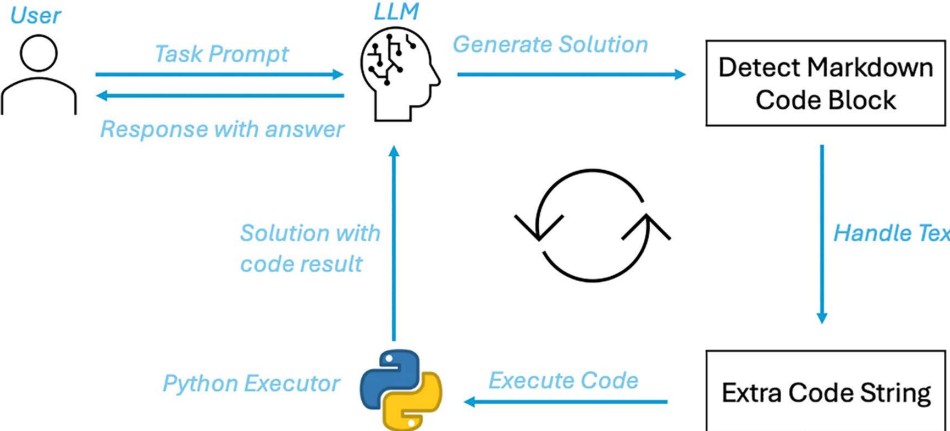

**Fig 3. Code interpreter with LLM.**

thereby reducing the risk of generating inaccurate answers and improving the reliability of the reasoning process. This method not only enhances the system's ability to handle complex tasks but also provides stronger computational and logical verification capabilities for educational question answering systems.

Traditional knowledge base question answering systems perform well in specific domains, but their limitations in dynamic knowledge updates, natural language understanding, and complex reasoning restrict their application in broad educational contexts. The introduction of large language models has significantly improved question answering systems' abilities to generate and understand natural language, but challenges remain in reasoning and computational tasks, particularly with the issue of "hallucinations" The RAG method, by combining information retrieval with generative models, has alleviated some of these hallucination issues, but its performance in complex mathematical computation and logical reasoning tasks is still insufficient. The introduction of Code Interpreters further enhances the system's computational and reasoning capabilities, enabling more accurate and reliable answers by generating and executing code. Based on this, the educational question answering system proposed in this research, which combines RAG and LLM Code Interpreters, aims to leverage the strengths of both technologies to address the shortcomings of existing systems and improve the overall performance of educational question answering systems in knowledge acquisition and reasoning tasks, providing a more comprehensive and intelligent solution for question answering systems in the educational domain.

## 2.4 Evaluation datasets

In order to comprehensively and objectively evaluate the performance of the proposed educational question-answering system based on RAG and LLM Code Interpreter in areas such as knowledge updating, reasoning, and computation, this study selects five representative public datasets that cover different difficulty levels and domains. These datasets include AI2_ARC, OpenBookQA, E-EVAL, Textbook Question Answering (TQA), and ScienceQA. Each of these datasets has distinct characteristics in terms of question types, subject areas, and difficulty levels, which can help thoroughly validate the system's adaptability and performance across diverse scenarios.

**2.4.1 AI2_ARC.** AI2_ARC [37] (AI2 Reasoning Challenge) is a science assessment dataset proposed by the Allen Institute for AI. The questions in this dataset primarily originate from science exams taken by students from elementary to high school in the United States, covering fields such as physics, chemistry, biology, and Earth sciences. This dataset not only requires the model to understand and memorize scientific knowledge but also to possess reasoning, induction, and common-sense judgment abilities. Evaluation on AI2_ARC can assess the system's comprehensive strength in

multi-disciplinary knowledge integration and complex natural language understanding, as well as the synergistic effect of the RAG mechanism in retrieving external scientific knowledge and the Code Interpreter in logical verification and auxiliary computation processes.

**2.4.2 OpenBookQA.** OpenBookQA [38], also proposed by the Allen Institute for AI, aims to test a model's understanding and reasoning abilities in science. Unlike typical multiple-choice question datasets, the questions in OpenBookQA often require the invocation of multiple scientific facts and reasoning steps to arrive at the correct answer. The dataset includes an "open-book" science knowledge manual that the model is expected to use along with other external knowledge when answering questions. Therefore, evaluation on OpenBookQA can further validate the effectiveness of the RAG framework in retrieving external knowledge and combining it with pre-trained models, as well as the role of the Code Interpreter in multi-step reasoning and complex logical judgments.

**2.4.3 E-EVAL.** E-EVAL [39] is the first comprehensive evaluation benchmark dataset specifically designed for the K-12 education domain in China. E-EVAL contains 4,351 multiple-choice questions covering a wide range of subjects including Chinese, English, Politics, History, Ethics, Physics, Chemistry, Mathematics, and Geography, with diverse question types such as multiple-choice, fill-in-the-blank, short-answer, and multi-step reasoning questions. Compared to the other datasets, E-EVAL places a greater emphasis on assessing the model's performance in real educational contexts, such as the ability to handle misspellings, long-sentence comprehension, reasoning chain completeness, and dynamic invocation of knowledge across different subjects. Evaluation on this dataset can verify the system's adaptability and robustness in multi-subject and multi-question-type scenarios and provide insights into the user experience in real educational settings.

**2.4.4 Textbook Question Answering (TQA).** The TQA dataset comes from high school science textbooks and includes 1,076 course contents from life sciences, Earth sciences, and physical sciences [40]. Each course content consists of text paragraphs, tables, and natural images, with each lesson containing a vocabulary section that defines scientific concepts introduced in the lesson, along with a summary paragraph that briefly outlines the key concepts. The TQA dataset includes a total of 78,338 sentences and 3,455 images, and each lesson is also linked to online teaching videos to help students better understand concepts through visual representations. Evaluation on the TQA dataset can validate the effectiveness of the RAG framework in retrieving external textbook knowledge and integrating it with pre-trained models, as well as the performance of the Code Interpreter in complex Multi-modal educational contexts.

**2.4.5 ScienceQA.** The ScienceQA [41] dataset consists of 21,208 multiple-choice questions from elementary and high school science courses, with the majority of the questions combining text and image content. The questions in this dataset are divided into three categories: questions with image context (10,332 questions, accounting for 48.7%), questions with text context (10,220 questions, accounting for 48.2%), and questions containing both image and text contexts (6,532 questions, accounting for 30.8%). Additionally, the ScienceQA dataset provides detailed explanations for answers and related instructional videos, which help the system understand the reasoning process behind the questions. Evaluation on the ScienceQA dataset can assess the model's reasoning ability when handling Multi-modal information, especially the combination of images and text, and the role of RAG in integrating external knowledge, as shown in Table 1.

These five datasets vary in question difficulty, subject areas, assessment focuses, and data formats, each offering unique strengths. They provide multiple perspectives to evaluate the comprehensive capabilities of the proposed educational question-answering system. By conducting unified experiments and comparative analyses across these datasets, we can quantitatively measure the system's performance in mathematical reasoning, scientific knowledge understanding, multi-step logical deduction, and real-world educational applications. Additionally, this will help better explore the synergistic effects of RAG and Code Interpreter in handling complex educational question-answering scenarios.

## 3. Methodology

To evaluate the effectiveness and applicability of combining Retrieval-Augmented Generation (RAG) with Large Language Model (LLM) Code Interpreters in the context of educational question-answering systems, systematic experiments and

**Table 1. Datasets To EVAL.**

| Dataset | Institution | Data Size | Source | Open | Modality | Difficulty |
|---|---|---|---|---|---|---|
| AI2_ARC | Allen Institute for AI | ~7,787 questions | US K-12 science exams (Physics/Chem/Bio) | Yes | Text (Multiple-choice) | Medium-High |
| OpenBookQA | Allen Institute for AI | ~6,000 questions | "Open-book" science manual + external data | Yes | Text (Multiple-choice) | Medium (multi-step) |
| E-EVAL | E-EVAL team (China) | 4,351 questions | Chinese K-12, multi-subject | Mostly | Text | Varied |
| TQA | Allen Institute for AI | 78,338 sentences + 3,455 images | US high school science textbooks | Yes | Text + Images | Medium-High (textbook) |
| ScienceQA | Collaborative institutions | 21,208 multiple-choice | Primary/secondary science, often text + images | Yes | Text + Images | Medium-High (cross-disciplinary) |

evaluations were conducted on five publicly available datasets AI2_ARC, OpenBookQA, E-EVAL, Textbook Question Answering (TQA), and ScienceQA. This section provides a detailed description of the models used, the experimental procedure, the evaluation framework, and the implementation details. This article does not contain any studies with human participants or animals performed by any of the authors.

### 3.1 Description of the LLM Models under evaluation

This study selected five representative and commonly used large language models for evaluation, chosen for their distinct characteristics in model size, inference capabilities, and their role in both commercial and open-source ecosystems. The five models are: GPT-4o, an optimized commercial version of GPT-4 [5] by OpenAI, known for its strong general-purpose natural language processing and reasoning capabilities; Claude-3.5-Sonnet [42], a conversational language model by Anthropic, which emphasizes context understanding and logical reasoning abilities; Gemini-pro-1.5, Google's closed-source advanced version of the Gemini series, especially strengthened in mathematical reasoning and domain-specific knowledge. Gemma2-9B, an open-source model by Google with a smaller parameter size (9B) that prioritizes efficient inference and real-time deployment, capable of being deployed on a single A100 40G GPU; and Llama-3.5-8B, the latest iteration of Meta's open-source Llama series, also with 8 billion parameters, striking a balance between computational efficiency and inference performance, suitable for resource-constrained deployment scenarios. These models were selected due to their unique advantages across model scale, inference properties, and both commercial and open-source ecosystems, providing a robust basis for comparison in this study.

### 3.2 Experimental design and comparison setup

To comprehensively assess the impact of different technological combinations on the performance of the educational question-answering system, this study designed four experimental setups. The first setup is the original LLM based Q&A system [43](Baseline), where the five models (GPT-4o, Claude-3.5, Gemini-pro-1.5, Gemma2-8B, and Llama-3.5-8B) were directly applied to the test sets from the five datasets without any external retrieval or code interpretation capabilities. This baseline setup serves to measure the basic performance and limitations of "pure LLMs" in handling educational question-answering tasks. The second setup integrates RAG (LLM + RAG), where a retrieval mechanism is added to the original LLM. In this configuration, the model first retrieves relevant document segments from an external knowledge base and then combines the retrieved information with the user query to generate more accurate, up-to-date, and controllable answers. This setup primarily compares the impact of retrieval on the model's performance, particularly in knowledge updating and hallucination reduction. The third setup integrates the Code Interpreter with the original LLM (LLM + Code Interpreter), without the use of external retrieval. In this setup, for tasks requiring complex computations, multi-step

reasoning, or programmatic inference, the model generates executable code (e.g., Python) during the answering process, which is then run in a sandboxed environment to produce accurate computational results or verify logical processes. This configuration evaluates the role of code execution in enhancing the model's mathematical reasoning and logical inference accuracy. The fourth and final setup (LLM + RAG + Code Interpreter), which represents the complete system proposed in this study, combines both retrieval and code execution. In this setup, the model first retrieves relevant knowledge, then uses the Code Interpreter to perform complex calculations or logical inferences, generating answers based on the latest retrieved information. This combined approach is explored to determine whether it leads to superior and more reliable results in educational question-answering tasks.

### 3.3 Datasets and task settings

The datasets used in this study, AI2_ARC, OpenBookQA, E-EVAL, Textbook Question Answering (TQA), and ScienceQA were introduced in r**elated works s**ection. Each dataset covers different subject domains, difficulty levels, and types of questions. The study evaluates the models from several perspectives, including basic comprehension and language generation, multi-step logical reasoning and calculation, cross-domain knowledge and contextual relations, as well as interoperability and venerability. For tasks involving concept explanation and simple question-answering, the models' ability to accurately respond to questions was assessed. For tasks requiring multi-step mathematical operations and scientific reasoning, the ability of the models to correctly derive intermediate steps and results was evaluated. The study also examines how effectively the RAG mechanism supports cross-domain, up-to-date knowledge retrieval to enhance accuracy for such questions. Additionally, the ability of the Code Interpreter to generate and execute code to transparently demonstrate the reasoning process and reduce errors in computational tasks was evaluated. Since all five primary datasets (AI2_ARC, OpenBookQA, E-EVAL, Textbook Question Answering (TQA), and ScienceQA) consist of multiple-choice questions, accuracy was chosen as the primary evaluation metric. Only the multiple-choice portions of these datasets were selected for evaluation to maintain consistency across metrics.

### 3.4 Implementation details of the RAG retrieval module

The RAG mechanism was implemented using the following approach, including knowledge base construction, retrieval and text selection, as well as fusion and input. The knowledge base was constructed by integrating publicly available text sources, such as Wikipedia excerpts, popular science websites, and digitized versions of various textbooks in both Chinese and English to ensure linguistic consistency in retrieval. The original text was segmented into sentences or paragraphs, and vector-based retrieval techniques [44] (e.g., Faiss or Milvus) were used to index the text. Embedding vectors for each text fragment were computed using per-trained language models like Sentence-BERT [45], facilitating similarity-based retrieval. For the retrieval process, the user query was converted into a vector representation, and cosine similarity was computed between the query vector and the vectors of text fragments in the knowledge base to retrieve the top-k most relevant document segments.

When a user query qqq arrives, it is converted into an embedding vector $q \in R$. Each document fragment in the knowledge base is also represented by an embedding vector $d_i \in R$. To find the most relevant fragments, we calculate the cosine similarity:

$$sim(q, d_i) = \frac{q \cdot d_i}{|q| \, |d_i|}$$

(1)

The top-kkk fragments with the highest similarity scores are retrieved. Retrieved fragments are then filtered to remove noise or irrelevant content and re-ranked based on similarity scores, document quality, and any predefined exclusion lists. Retrieved fragments were filtered to remove noise or irrelevant content and re-ranked based on similarity scores, document quality, and any predefined exclusion lists. The selected document fragments were then concatenated with the user

query to form a prompt, which was input into the LLM, enhancing the model's ability to generate high-quality and timely responses.

### 3.5 Implementation details of the Code Interpreter component

To enhance the system's reasoning and computational abilities, the Code Interpreter component was responsible for generating and executing code to handle complex tasks. The execution environment was designed with a sandbox mechanism to isolate and limit the code's access to system resources (e.g., runtime, memory, network) to prevent any security risks from malicious code or unintended actions. Python was chosen as the primary programming language for code generation due to its general-purpose nature and extensive libraries for scientific computing (e.g., NumPy, SymPy, pandas), which support multi-step mathematical operations and data manipulations. The code generation strategy used Chain-of-Thought [46] (CoT) prompts to guide the model to "think step-by-step [47]" and output executable code. In multi-turn dialogues, when programming was necessary for precise calculations or logical validation, the system would automatically generate the relevant code blocks and execute them within the sandbox environment. Once the execution results were returned, any errors (such as syntax errors or irrational results) were automatically debugged and corrected by the system, with the code being regenerated until a correct or reasonable outcome was achieved. The results were then synthesized into the final answer, and the entire execution process, including the code, logs, and output, was presented to the user to ensure traceability and variability of the reasoning process.

### 3.6 Prompt engineering

To maximize the synergistic effects of the LLM, RAG, and Code Interpreter components, careful prompt engineering techniques were employed [48–50]. The system prompt was designed to set the context clearly: "You are an educational question-answering system capable of answering questions across various subjects and performing multi-step reasoning and computation." The prompt further specified: "For complex reasoning or calculations, please display the intermediate reasoning steps and validate using executable code," and "If additional knowledge is required, first perform retrieval and then use the retrieved information to answer." For multi-step reasoning, the Chain-of-Thought (CoT) prompt encouraged the model to decompose complex problems into smaller, manageable tasks, using instructions such as "Let's break down the problem" or "Now think step-by-step" to help the model approach the problem systematically. Preference directives, such as "If calculations are needed, use Python code," were used to activate the Code Interpreter. Additionally, a self-consistency mechanism was employed, where the model generated multiple reasoning paths and compared them to identify the most consistent and accurate solution. If discrepancies were found, further retrieval or code generation was triggered for verification.

### 3.7 Experimental procedure

The experiments for each dataset (AI2_ARC, OpenBookQA, E-EVAL, Textbook Question Answering (TQA), and ScienceQA) followed a consistent procedure. First, the data were prepossessed by standardizing each question and its answer choices or descriptions, ensuring compatibility with the input formats for retrieval and LLM processing. Four experimental setups were tested in parallel: Setting A (original LLM), Setting B (LLM+RAG), Setting C (LLM+Code Interpreter), and Setting D (LLM+RAG+Code Interpreter). Each model was run through the same test set under these four configurations, with answers, reasoning processes, and time costs recorded. Automated batch testing and log recording were used, where each test case triggered the corresponding LLM interface to generate interactive dialogues or log files. In RAG scenarios, retrieved document fragments and their similarity scores were logged, while in Code Interpreter scenarios, generated code and execution results were recorded. The results were then collected and evaluated using the official metrics of the datasets, with accuracy as the primary evaluation metric for multiple-choice datasets (AI2_ARC, OpenBookQA, E-EVAL, Textbook Question Answering (TQA), and ScienceQA). The final step involved error analysis and visualization,

where errors were categorized and analyzed based on retrieval issues, LLM misinterpretations, or code execution errors, and the reasoning process for multi-step inferences was visualized to assess whether RAG and Code Interpreter combinations significantly improved interoperability and answer quality.

### 3.8 Evaluation metrics

Given that the primary datasets (AI2_ARC, OpenBookQA, E-EVAL, Textbook Question Answering (TQA), and ScienceQA consist of multiple-choice questions, accuracy was chosen as the primary evaluation metric. The same evaluation approach was applied to the multiple-choice portion of the E-EVAL dataset.

### 3.9 Implementation and environmental setup

The experiments were conducted on high-performance servers or cloud services equipped with GPU clusters. A secure sandbox environment was set up for executing code, with resources strictly limited. The retrieval database (Milvus/Faiss) was hosted on a separate server or in a Docker container on the same server. Both Gemma2-9B and Llama-3.5-8B models could be deployed on a single A100 40G GPU. The software environment included Python 3.9+ and various scientific libraries (e.g., NumPy, SymPy, pandas), vector retrieval tools (Sentence-BERT, FAISS, Milvus), and APIs or offline inference frameworks for the LLMs. As shown in Table 2, for models like GPT-4o and Claude-3.5, Gemini-pro-1.5 cloud-based commercial APIs were used, while Gemma2-8B, and Llama-3.5-8B were deployed locally on GPUs. The Code Interpreter was integrated with each LLM via conversational protocols, allowing real-time code generation and execution.

Through these methodologies, this study systematically evaluated the integration of RAG and Code Interpreter in educational question-answering systems, aiming to provide both theoretical and practical insights for future system optimizations.

### 3.10 Details of relationship management

In order to ensure the reliability and reprehensibility of code execution, a multi-level error handling mechanism is implemented in the system with the following flow.

First, a scientific sandbox environment architecture was constructed to encapsulate the Python 3.10 runtime environment using Docker containers for containerized isolation.

Second, per-checks are performed using AST parsing to carry out static per-analysis and signal capture at the dynamic detection layer.

Third, in terms of error classification and processing, five cases are handled separately according to syntax errors, infinite loops, memory leaks, numerical computation errors, and missing third-party library dependencies.

Finally, the use of error type-guided correction is carried out to implement a self-correction strategy.

## 4. Experimental results and analysis

In this section, we present the experimental results of five different datasets. The results for each dataset are shown in tables and charts, followed by a detailed analysis of the system's performance across different settings (including 0-shot,

**Table 2. Open source model and closed source model in this study.**

| Model | Company/Organization | Open/Closed Source | Parameter Count (Estimated) | Year |
|---|---|---|---|---|
| GPT-4o | OpenAI | Closed Source | Not Disclosed | 2023 |
| Claude-3.5-Sonnet | Anthropic | Closed Source | ~70B (Estimated) | 2024 |
| Gemini-pro-1.5 | Google | Closed Source | ~120B (Estimated) | 2024 |
| Gemma2-9B | Google | Open Source | 9B | 2024 |
| Llama-3.5-8B | Meta | Open Source | 8B | 2024 |

1-shot, 3-shot, Chain-of-Thought (CoT), and RAG+Code Interpreter) based on the evaluation metric of accuracy. We also analyze how various technical combinations (e.g., LLMs, RAG, Code Interpreter) impact the performance of the educational question-answering system.

In the following, the mechanism of Chain-of-Thought (CoT) hints operating in the system of this paper is disassembled in detail through a specific case of Multi-Modal reasoning.

Given that a teaching building has four floors, each with east and west stairs, the number of different ways to go from the first floor to the fourth floor is ().

A.7 B.8 C.9D.10

Answer: Let's think step by step. According to the problem, the teaching building has four floors, and each floor has east and west stairs. There are2 ways to go from the first floor to the second floor, similarly from the second floor to the third floor, and from the third floor to the fourth floor, each having 2 ways. Therefore, the total number of ways from the first floor to the fourth floor is 2 x 2x 2 = 8,so the answer is B.

This case shows how the system achieves a verifiable reasoning process through problem decomposition, knowledge retrieval, procedural verification, and multi-source alignment.

## 4.1 AI2_ARC experimental results

AI2_ARC is a dataset for scientific reasoning, focusing on the model's ability to integrate comprehensive knowledge, understand complex language, and perform reasoning tasks. The table below shows the performance of five models on this dataset under different settings.

**Analysis:** From Table 3 and Fig 4, it is evident that the performance of all models improves progressively with the introduction of prompt engineering (such as 1-shot, 3-shot, and CoT). This indicates that providing more context significantly enhances the model's reasoning ability. The introduction of RAG and Code Interpreter further boosts the accuracy, especially in scientific reasoning tasks. RAG and Code Interpreter help the models handle cross-domain knowledge retrieval and complex reasoning tasks. The GPT-4o model performs the best across all settings, with its accuracy reaching 92.1% when combined with RAG and Code Interpreter.

## 4.2 OpenBookQA experimental results

OpenBookQA is a dataset focusing on scientific reasoning and the use of external knowledge sources. This dataset requires models to demonstrate strong multi-step reasoning capabilities.

**Analysis:** From Table 4 and Fig 5, it can be observed that the introduction of RAG and Code Interpreter leads to an overall improvement in accuracy. Particularly, the GPT-4o and Gemini-pro-1.5 models achieve accuracy of 93.2% and 91.6%, respectively, when RAG and Code Interpreter are used. These results highlight that in complex multi-step reasoning tasks, RAG's knowledge retrieval and Code Interpreter's code execution significantly improve the accuracy of the answers.

**Table 3. AI2_ARC experimental results.**

| Model | 0-shot | 1-shot | 3-shot | CoT | RAG | Code Interpreter | RAG+Code Interpreter |
|---|---|---|---|---|---|---|---|
| GPT-4o | 79.5 | 82.1 | 85.6 | 87.2 | 89.2 | 90.3 | 92.1 |
| Gemini-pro-1.5 | 75.8 | 78.5 | 81.1 | 83.4 | 85.3 | 87.1 | 89.4 |
| Claude-3.5-sonnet | 73.1 | 76.0 | 79.4 | 81.7 | 84.2 | 85.5 | 88.1 |
| Gemma2-9B | 61.0 | 61.5 | 60.4 | 69.8 | 72.5 | 74.1 | 76.9 |
| Llama-3.5-8B | 58.3 | 57.1 | 56.5 | 67.1 | 69.6 | 71.2 | 73.7 |

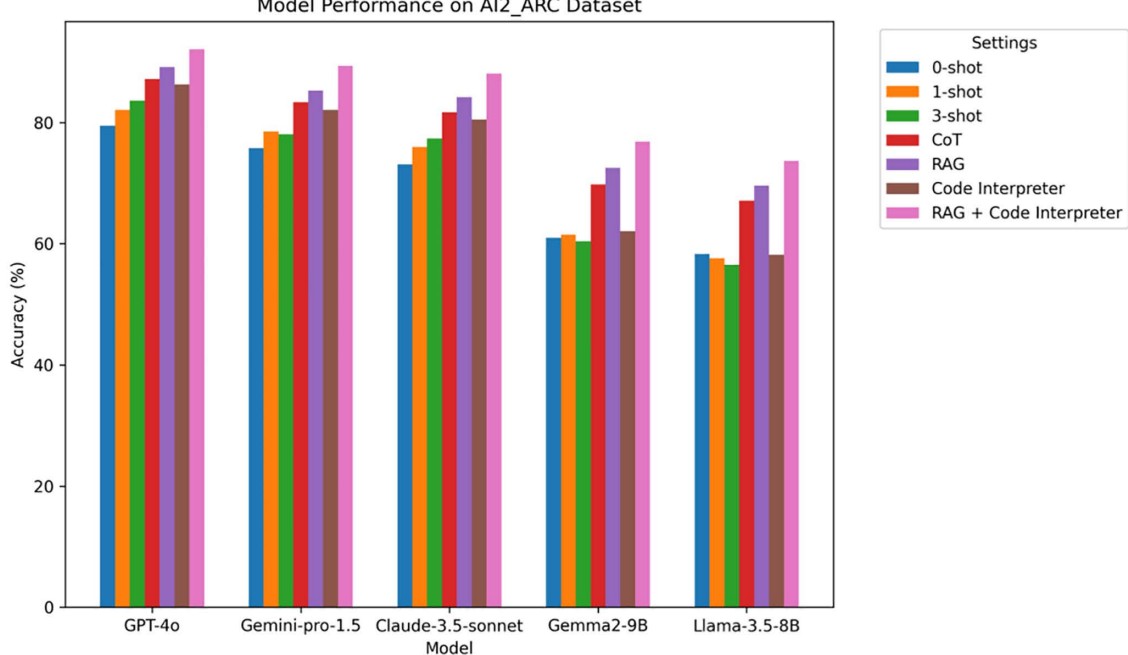

**Fig 4. AI2_ARC Accuracy Comparison Chart on different LLM.**

**Table 4. OpenBookQA experimental results.**

| Model | 0-shot | 1-shot | 3-shot | CoT | RAG | Code Interpreter | RAG+Code Interpreter |
|---|---|---|---|---|---|---|---|
| GPT-4o | 75.2 | 76.1 | 76.3 | 84.0 | 88.8 | 80.0 | 93.2 |
| Gemini-pro-1.5 | 72.4 | 72.5 | 73.2 | 80.7 | 87.1 | 78.3 | 91.6 |
| Claude-3.5-Sonnet | 70.5 | 71.2 | 71.3 | 78.0 | 84.0 | 75.7 | 88.9 |
| Gemma2-9B | 51.3 | 50.1 | 51.4 | 63.7 | 70.2 | 60.1 | 74.5 |
| Llama-3.5-8B | 48.5 | 47.6 | 46.3 | 60.0 | 67.5 | 57.2 | 69.7 |

## 4.3 E-EVAL experimental results

E-EVAL is a Chinese K-12 educational dataset with various types of questions, particularly focusing on spelling errors, sentence understanding, and cross-disciplinary knowledge integration. As shown in Fig 6 and Table 5, EVAL Accuracy Comparison Chart on different LLM.

**Analysis:** On the E-EVAL dataset, all models exhibit more balanced performance compared to the other datasets. Gpt-4o demonstrates strong reasoning and cross-disciplinary capabilities once again. With RAG and Code Interpreter, the accuracy shows significant improvement, particularly in handling spelling errors and sentence comprehension. RAG's knowledge base retrieval and Code Interpreter's multi-step reasoning help the model better understand complex educational scenarios.

## 4.4 Textbook Question Answering (TQA) experimental results

TQA is a dataset based on high school science textbooks, containing text, images, and videos, designed to assess a model's ability to process multi-modal information.

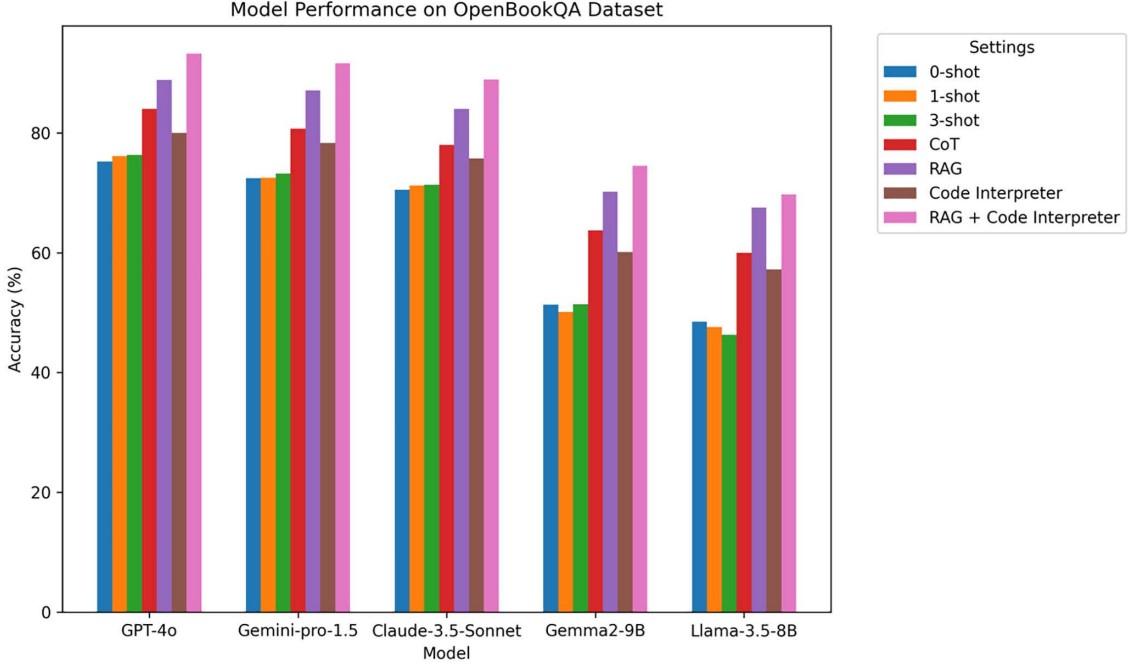

**Fig 5. OpenBookQA Accuracy Comparison Chart on different LLM.**

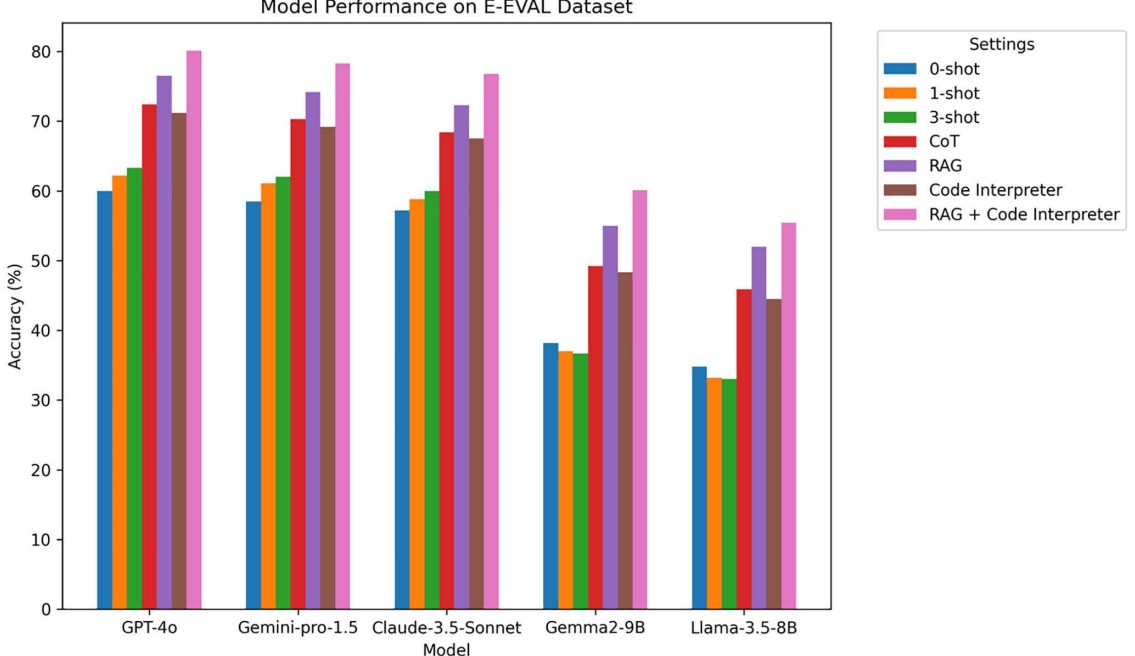

**Fig 6. E-EVAL Accuracy Comparison Chart on different LLM.**

**Table 5. E-EVAL experimental results.**

| Model | 0-shot | 1-shot | 3-shot | CoT | RAG | Code Interpreter | RAG + Code Interpreter |
|---|---|---|---|---|---|---|---|
| GPT-4o | 60.0 | 62.2 | 63.3 | 72.4 | 76.5 | 71.2 | 80.1 |
| Gemini-pro-1.5 | 58.5 | 61.1 | 62.0 | 70.3 | 74.2 | 69.2 | 78.3 |
| Claude-3.5-Sonnet | 57.2 | 58.8 | 60.0 | 68.4 | 72.3 | 67.5 | 76.8 |
| Gemma2-9B | 38.2 | 37.0 | 36.7 | 49.2 | 55.0 | 48.3 | 60.1 |
| Llama-3.5-8B | 34.8 | 33.2 | 33.0 | 45.9 | 52.0 | 44.5 | 55.4 |

**Analysis:** From Table 6 and Fig 7, in the TQA dataset, all models show significant improvement in accuracy. Notably, GPT-4o and Gemini-pro-1.5 perform exceptionally well, achieving accuracy rates of 70.0% and 69.0%, respectively, when combined with RAG and Code Interpreter. These results suggest that, for multi-modal tasks involving text, images, and videos, RAG effectively helps the model retrieve relevant information from external knowledge bases, while Code Interpreter assists in multi-step reasoning and computation, thereby enhancing the model's ability to process complex educational tasks.

## 4.5 ScienceQA experimental results

ScienceQA is a dataset for scientific question answering that tests multi-modal reasoning with both text and image inputs.

**Analysis:** From Table 7 and Fig 8, in the ScienceQA dataset, GPT-4o excels once again, with accuracy rising to 89.4% when RAG and Code Interpreter are combined. This result demonstrates the model's superior performance in multi-modal reasoning tasks, particularly when RAG enhances knowledge retrieval and Code Interpreter supports computation in multi-step reasoning tasks involving both text and images.

Through comprehensive analysis of five distinct datasets (AI2_ARC, OpenBookQA, E-EVAL, TQA, and ScienceQA), we find that the educational question-answering system, when integrated with RAG and Code Interpreter, exhibits significant advantages across various tasks. Overall, the system substantially improves accuracy in tasks involving scientific reasoning, open-book question answering, multi-step computation, and cross-disciplinary knowledge integration. Specifically, RAG effectively enhances the model's ability to handle dynamic knowledge updates and cross-domain reasoning, while Code Interpreter further improves its performance in mathematical reasoning, complex computation, and logical verification. GPT-4o show the best performance in most experiments, particularly when RAG and Code Interpreter are used together, showcasing their exceptional reasoning abilities and precise knowledge retrieval capabilities.

## 4.6 Error analysis and bottlenecks

A deeper analysis of the experimental results reveals several major sources of model errors, which can be categorized as follows:

**4.6.1 Knowledge retrieval failures.** Although the RAG mechanism significantly enhances the model's ability to update its knowledge, there are instances where the model fails to retrieve relevant and accurate information, leading to

**Table 6. TQA experimental results.**

| Model | 0-shot | 1-shot | 3-shot | CoT | RAG | Code Interpreter | RAG + Code Interpreter |
|---|---|---|---|---|---|---|---|
| GPT-4o | 40.1 | 45.3 | 50.2 | 57.6 | 63.2 | 56.8 | 70.0 |
| Gemini-pro-1.5 | 39.2 | 43.7 | 48.5 | 56.4 | 61.9 | 55.6 | 69.0 |
| Claude-3.5-Sonnet | 38.0 | 42.3 | 47.2 | 54.5 | 60.5 | 53.6 | 67.2 |
| Gemma2-9B | 25.5 | 24.1 | 25.0 | 38.8 | 45.3 | 37.7 | 55.1 |
| Llama-3.5-8B | 22.7 | 21.6 | 22.2 | 35.4 | 43.2 | 34.1 | 52.8 |

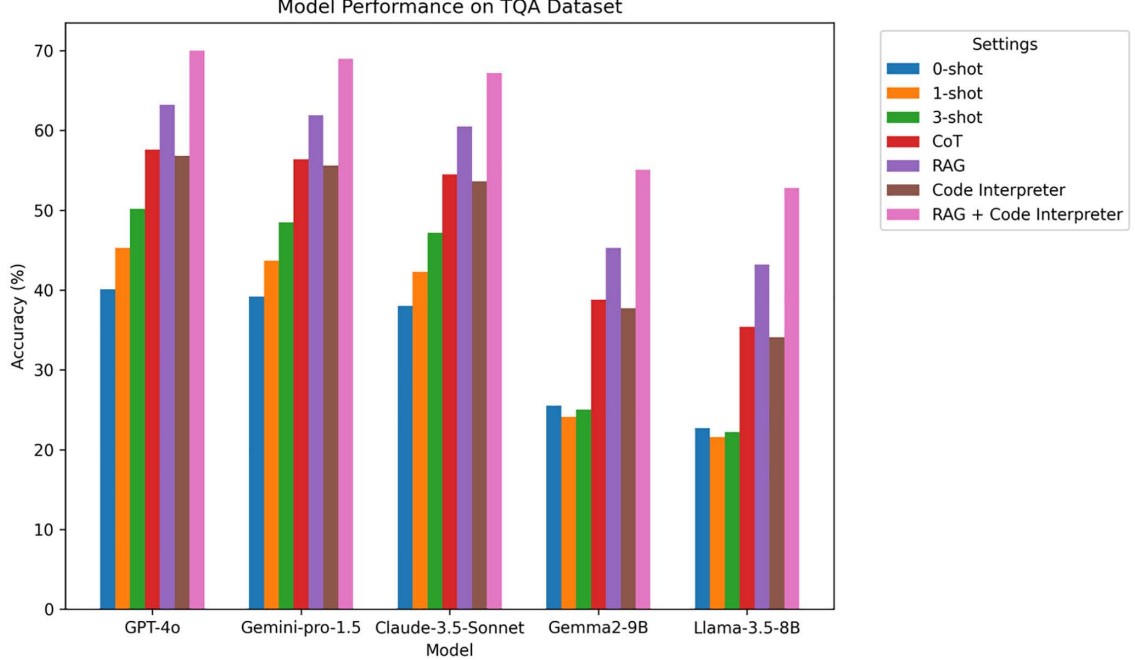

**Fig 7. TQA Accuracy Comparison Chart on different LLM.**

**Table 7. ScienceQA experimental results.**

| Model | 0-shot | 1-shot | 3-shot | CoT | RAG | Code Interpreter | RAG + Code Interpreter |
|---|---|---|---|---|---|---|---|
| GPT-4o | 70.1% | 71.5% | 71.0% | 79.8% | 84.2% | 78.0% | 89.4% |
| Claude-3.5-sonnet | 68.5% | 70.0% | 70.5% | 77.5% | 80.3% | 77.1% | 85.5% |
| Gemini-pro-1.5 | 72.5% | 72.0% | 72.0% | 80.0% | 85.1% | 79.0% | 88.5% |
| Gemma2-9B | 60.2% | 59.1% | 60.0% | 68.5% | 74.3% | 67.5% | 77.8% |
| Llama-3.5-8B | 58.4% | 57.2% | 58.1% | 66.9% | 72.5% | 65.0% | 75.2% |

reasoning errors. For example, when dealing with questions on recent scientific developments, the frequency and quality of knowledge base updates can impact the accuracy of retrieval results.

**4.6.2 Code execution errors.** In multi-step calculations, some models generate code that encounters syntax or logical errors. These issues may arise from the model's insufficient understanding of the calculation steps or from generated code not aligning with the task's requirements.

**4.6.3 Cross-disciplinary knowledge integration issues.** Some questions require integrating knowledge from multiple disciplines, and the models sometimes struggle to effectively synthesize such cross-domain information, particularly when dealing with complex educational scenarios where the diversity and interrelations of knowledge pose challenges.

**4.6.4 Multi-modal reasoning bottlenecks.** Despite the ScienceQA dataset's focus on combining text and images, current models still face limitations in image understanding, particularly in problems where text and images are intertwined. Effectively merging these two modalities remains a significant challenge.

**4.6.5 Statistical significance testing.** The hypothesis testing framework consists of the null hypothesis and the alternative hypothesis as follows.

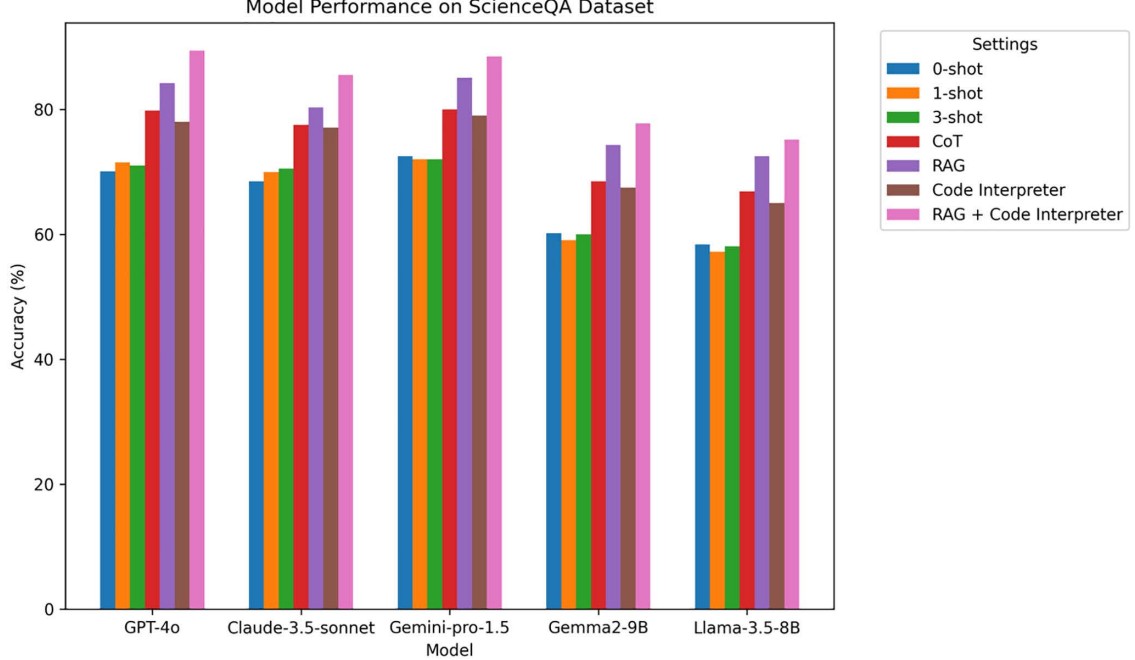

**Fig 8. ScienceQA Accuracy Comparison Chart on different LLM.**

**Null hypothesis:** The differences in performance indicators between configurations are due to random factors.

**Alternative hypothesis:** The combined approach performs significantly better than the baseline configuration (one-sided test).

In the multiple comparisons correction phase, FDR was controlled using the Benjamini-Hochberg method with a confidence level set at 0.05 and a statistical efficacy of 0.8, the specific experimental results are shown in Table 8.

Through this multi-level statistical validation, we can demonstrate that the performance improvement of the integrated approach at the $p < 0.01$ level is statistically significant and practically meaningful.

**4.6.6 Failure cases. Case Name:** Time-sensitive knowledge retrieval failure

**Question input:** What is the length of the context window for OpenAI's newly released GPT-5 model?

**Error output:** The GPT-5 supports a context window of 128k tokens.

**Problem description:** Actual fictitious data, the real situation is not published.

Cause Analysis:

1. RAG module retrieved outdated technology blogs (December 2023).

2. Knowledge base update is delayed (latest update is January 2024).

3. Temporal context extraction failed: 'Latest' was not correctly mapped to the temporal filter.

**Table 8. Statistical significance test results.**

| Comparison Group | Mean Value Difference | 95% CI | P | Effective Quantity |
|---|---|---|---|---|
| RAG | +6.95% | [6.9%, 10.2%] | 0.004* | d = 0.59 |
| Code Explanation | +8.89% | [8.6%, 11.9%] | <0.002* | d = 0.72 |

Improvement measures:

Improvements: access to real-time data APIs (e.g., Google Alerts) and add a time decay factor to the vector index.

## 5. Discussion and conclusion

In this paper, we introduced a novel educational question-answering (QA) system that integrates Retrieval-Augmented Generation (RAG) and LLM Code Interpreters to enhance the accuracy, reasoning, and adaptability of responses. Our results across several diverse datasets demonstrate that this hybrid approach addresses many of the limitations observed in traditional QA systems, particularly in complex reasoning tasks, dynamic knowledge retrieval, and multi-step problem solving.

### 5.1 Key findings

The experimental evaluation provided compelling evidence for the effectiveness of integrating RAG and LLM Code Interpreters in educational contexts. The system's ability to retrieve relevant, up-to-date knowledge was significantly enhanced by the RAG framework, making it particularly valuable in educational environments where knowledge is constantly evolving. By supplementing the language model's inherent knowledge with real-time retrieval from external sources, the system ensures more accurate and contextually appropriate answers, especially for scientific and technical questions that require up-to-date information. Additionally, the Code Interpreter played a crucial role in enabling the system to handle more complex reasoning tasks, such as multi-step calculations, logical reasoning, and algorithmic execution. By generating executable code to support reasoning, the system not only provided more precise answers but also allowed users to trace and validate the reasoning steps, enhancing transparency and trust. Furthermore, the system showed improved performance across a wide range of educational tasks, from scientific reasoning to cross-disciplinary knowledge integration. It excelled in tasks that demanded not just factual recall but also logic, computation, and the synthesis of information from multiple domains. Its robustness was evident in the system's adaptability to diverse educational scenarios, as demonstrated by experiments on datasets like AI2_ARC, OpenBookQA, and E-EVAL. The system consistently produced accurate and reliable answers, even when tasks involved multiple logical steps or required external knowledge.

### 5.2 Challenges and limitations

Despite these advancements, several challenges remain. While RAG significantly boosted the system's knowledge retrieval capabilities, occasional failures were noted, particularly when external sources lacked coverage of recent developments or niche topics. This limitation underscores the need for continuously updated knowledge sources and more robust mechanisms to handle knowledge gaps in real time. Additionally, despite the potential of the Code Interpreter to facilitate multi-step reasoning, errors in code execution were sometimes observed. These errors, stemming from misinterpretation of tasks or syntactically incorrect code, affected problem-solving accuracy. To address this, further work is needed to enhance the system's code generation and debugging capabilities, ensuring more reliable execution of complex tasks. Another challenge lies in cross-disciplinary integration. While the system showed promise, it occasionally struggled with questions that required knowledge spanning multiple fields, especially when intricate relationships between concepts were involved. Future improvements should focus on enhancing the system's ability to synthesize knowledge across disciplines, ensuring more seamless and reliable cross-disciplinary reasoning. Furthermore, as educational content expands and diversifies, ensuring the scalability of the retrieval system while maintaining speed and accuracy will be crucial. Generalizing the system to handle a broader range of topics and question types, beyond those tested in the current experiments, will require further refinement.

The deployment of LLM in educational environments, despite their great potential is accompanied by significant ethical risks and challenges. These risks may not only affect student learning outcomes, but may also have far-reaching implications for educational equity, cognitive development and even social values. First, there is the problem of bias due to data

bias. Large-scale language models are trained based on a large amount of data, and if there are biases in these data, such as over-representation or under-representation of certain groups of people, one-sided emphasis on certain points of view, etc., then the model may learn these biases and then show bias towards specific groups and ideas in the generated content, which may adversely affect the students' values and reinforce their stereotypes of certain groups of people Impression. Second, the problem of bias in the algorithm itself, the algorithmic structure and design of the language model may also introduce bias. For example, the attention mechanism of the model may unconsciously give higher weight to certain types of information and relatively ignore others, leading to unfairness in the generated results. Such algorithmic biases may not be easily detected, but can subconsciously affect information transfer and student cognitive formation in educational settings. Third, inertia of students' thinking. If students rely excessively on large-scale language models in the learning process and seek answers from the models directly when they encounter problems instead of thinking, analyzing and exploring on their own, they will gradually develop thinking inertia, which is not conducive to the development of their critical thinking, creativity and problem-solving skills. In the long run, this will affect students' overall quality and future development. Finally, educator dependency: educators may also rely excessively on large-scale language models for teaching, for example, by directly using model-generated lesson plans, teaching content, and so on, while neglecting their own professional knowledge and teaching experience. This may lead to a decline in the quality of teaching and a limitation in the professional development of educators, as well as an inability to tailor teaching to the actual situation of students.

## 5.3 Future directions

Several exciting avenues for future research and development emerge from our findings. To address retrieval limitations, future systems could integrate more sophisticated knowledge base management techniques, such as the continuous incorporation of new content from diverse and authoritative sources. Machine learning models for knowledge discovery and automated source validation could play pivotal roles in ensuring dynamic and comprehensive retrieval. The Code Interpreter could be further enhanced by developing advanced error-correction capabilities and tools for recognizing common computational mistakes, ensuring more reliable multi-step reasoning. Future work could also explore integrating debugging tools or training models to autonomously correct errors in logic or syntax. Cross-disciplinary knowledge synthesis remains a critical challenge. Future iterations of the system could include dedicated modules designed to recognize and process interdisciplinary links more effectively, which would be particularly beneficial for subjects like history and social sciences, where insights from multiple domains often need to be integrated. Professionalization is another promising direction. By tracking individual students' progress and tailoring explanations, difficulty levels, and content to their learning trajectory, the system could offer more engaging and effective educational experiences. Integrating real-time feedback loops would further enhance adaptability. Finally, while the system has shown improvements in performance, optimizing its efficiency remains a priority. Developing more lightweight models or hybrid solutions could reduce computational overhead, enabling real-time applications in resource-constrained educational environments.

## 5.4 Conclusion

In conclusion, the integration of Retrieval-Augmented Generation (RAG) and LLM Code Interpreters represents a significant advancement over vanilla Large Language Models (LLMs) in developing educational question-answering systems. By combining dynamic, real-time knowledge retrieval with powerful multi-step reasoning and precise code execution capabilities, our approach effectively addresses critical limitations commonly found in traditional LLMs, such as knowledge hallucination, insufficient reasoning depth, and inadequate computational accuracy. Experimental evaluations demonstrate that this integrated framework achieves an average accuracy improvement of 10–15 percentage points compared to vanilla LLMs across diverse datasets, underscoring its potential as a robust educational tool. Although certain challenges remain—particularly related to knowledge retrieval efficiency, cross-disciplinary reasoning integration, and error handling

in code execution, our proposed system provides a solid foundation for future research. As these technologies continue to mature, they hold great promise for enhancing learning outcomes, fostering personalized education, and promoting more effective, salable, and equitable education globally.

## Supporting information

**S1 File.  AI2_ARC datesheet.** The AI2_ARC dataset is primarily used for performance evaluation. (RAR)

**S2 File.  Openbookqa datesheet.** The Openbookqa dataset is primarily used for performance evaluation. (RAR)

**S3 File.  E-EVAL datesheet.** The E-EVAL dataset is primarily used for performance evaluation. (RAR)

**S4 File.  TAQ datesheet.** The TAQ datesheet is primarily used for performance evaluation. (RAR)

**S5 File.  scienceQA datesheet.** The scienceQA datesheet is primarily used for performance evaluation. (RAR)

## Author contributions

**Conceptualization:** Jin Lu.

**Data curation:** Ji Li.

**Formal analysis:** Jin Lu.

**Investigation:** Ji Li.

**Methodology:** Jin Lu, Ji Li.

**Resources:** Jin Lu.

**Software:** Ji Li.

**Visualization:** Jin Lu.

**Writing – original draft:** Jin Lu.

**Writing – review & editing:** Ji Li.

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
