## [Decision Letter · Decision Letter 0]

25 Apr 2025

Dear Dr. Li,

Thank you for submitting your manuscript to PLOS One. After careful consideration, we feel that it has merit but does not fully meet PLOS One’s publication criteria as it currently stands. Therefore, we invite you to submit a revised version of the manuscript that addresses the points raised during the review process.

We look forward to receiving your revised manuscript.

Kind regards,

Hugh Cowley

Staff Editor

PLOS One

Journal Requirements:

“This work is partially supported by 2025 Key Research Project of Shenzhen Polytechnic University "Research on Key Methods for Analysis and Prediction of Social Behaviour of Specific Characters on Multimodal Big Data (6025310008K)", 2024 Guangdong Province Education Science Planning Project (Higher Education Special Project) "Research on Smart Classroom Teaching Behavior Analysis Methods Based on Scene Semantic Understanding and Deep Learning Characteristic Representation (2024GXJK766)", 2023 Guangdong Provincial Higher Vocational Education Teaching Reform Research and Practice Project "Research on Online Teaching Quality Evaluation Method Based on Multimodal Affective State Analysis (2023JG277)", 2024 Shenzhen Polytechnic University Quality Engineering Project "Research on Classroom Scene Understanding and Behavior Analysis Method Based on Multimodal Attention Mechanisms (7024310268)", 2024 Higher Education Scientific Research Planning Project of the Chinese Society of Higher Education "Research on the Analysis of Teaching and Learning Deep Interaction Characteristics in Smart Classroom Environment Supported by Multimodal Data (24XH0407)", 2023 Shenzhen Education Science Planning Project"Research on the Evolutionary Mechanism and Intervention of Interpersonal Relationships among College Students Driven by Multimodal Data (rgzn23003)".

“This work is partially supported by 2025 Key Research Project of Shenzhen Polytechnic University "Research on Key Methods for Analysis and Prediction of Social Behaviour of Specific Characters on Multimodal Big Data (6025310008K)", 2024 Guangdong Province Education Science Planning Project (Higher Education Special Project) "Research on Smart Classroom Teaching Behavior Analysis Methods Based on Scene Semantic Understanding and Deep Learning Characteristic Representation (2024GXJK766)", 2023 Guangdong Provincial Higher Vocational Education Teaching Reform Research and Practice Project "Research on Online Teaching Quality Evaluation Method Based on Multimodal Affective State Analysis (2023JG277)", 2024 Shenzhen Polytechnic University Quality Engineering Project "Research on Classroom Scene Understanding and Behavior Analysis Method Based on Multimodal Attention Mechanisms (7024310268)", 2024 Higher Education Scientific Research Planning Project of the Chinese Society of Higher Education "Research on the Analysis of Teaching and Learning Deep Interaction Characteristics in Smart Classroom Environment Supported by Multimodal Data (24XH0407)", 2023 Shenzhen Education Science Planning Project"Research on the Evolutionary Mechanism and Intervention of Interpersonal Relationships among College Students Driven by Multimodal Data (rgzn23003)".

5. We note that your Data Availability Statement is currently as follows: All relevant data are within the manuscript and its Supporting Information files.

6. PLOS requires an ORCID iD for the corresponding author in Editorial Manager on papers submitted after December 6th, 2016. Please ensure that you have an ORCID iD and that it is validated in Editorial Manager. To do this, go to ‘Update my Information’ (in the upper left-hand corner of the main menu), and click on the Fetch/Validate link next to the ORCID field. This will take you to the ORCID site and allow you to create a new iD or authenticate a pre-existing iD in Editorial Manager.

7. Please ensure that you refer to Figure 1, 3, 7, 8 and 9 in your text as, if accepted, production will need this reference to link the reader to the figure.

8. Please upload a copy of Figure 4, to which you refer in your text on page 19 in PDF submission. If the figure is no longer to be included as part of the submission please remove all reference to it within the text.

9. We note you have included a table to which you do not refer in the text of your manuscript. Please ensure that you refer to Table 1, 2, 3, 5, 6 and 7 in your text; if accepted, production will need this reference to link the reader to the Table.

Reviewers' comments:

Reviewer's Responses to Questions

**Comments to the Author**

1. Is the manuscript technically sound, and do the data support the conclusions?

Reviewer #1: Yes

Reviewer #2: Yes

2. Has the statistical analysis been performed appropriately and rigorously?

Reviewer #1: Yes

Reviewer #2: Yes

3. Have the authors made all data underlying the findings in their manuscript fully available?

Reviewer #1: Yes

Reviewer #2: Yes

4. Is the manuscript presented in an intelligible fashion and written in standard English?

Reviewer #1: Yes

Reviewer #2: Yes

Reviewer #1: In summary, this article is interesting and offers valuable findings in terms of both theoretical knowledge and practical application. However, there are several weaknesses that need to be addressed to enhance the overall quality of the article. Language errors are evident, and the writing style lacks consistency, particularly in the labelling of tables and figures. Some labels use lowercase letters while others capitalize each word, which disrupts uniformity. Additionally, the labels for Figures 1, 2, and 3 are unclear, and the colours used are not appropriate for academic writing.

The font used throughout the article is also inconsistent across the main text, references, and figure labels. For instance, the citation for “Figure 4” in the text does not follow a consistent formatting style. Moreover, it is recommended that Figure 4 be converted into a table, and all its content be incorporated accordingly for better clarity and presentation.

The presentation of research findings should also be further elaborated, with a more in-depth discussion that connects the current results to previous research findings and the theoretical framework underpinning the study.

Reviewer #2: Please consider correcting minor typographical errors such as “frameworkk” and “faik” to enhance the overall readability and polish of the manuscript.

It may be helpful to clarify the distinctions between the different experimental configurations (e.g., RAG only, Code Interpreter only, and the combined approach) earlier in the methodology section to guide readers through the experimental design more clearly.

Including a visual system architecture diagram that illustrates the interaction between the RAG module, the Code Interpreter, and the LLM could significantly aid in understanding the proposed framework.

To strengthen the rigor of the results, it is recommended to supplement the performance comparisons with appropriate statistical significance tests to demonstrate that the observed improvements are meaningful.

Providing additional detail on how the system handles or mitigates code execution errors—especially within the sandbox environment—would improve transparency and reproducibility.

It would be valuable to indicate whether any parameter tuning was performed for either the RAG or Code Interpreter components and, if so, describe the approach and settings used.

Expanding on the use of Chain-of-Thought prompting with a concrete example could help readers better appreciate how this strategy supports multi-step reasoning within the system.

The error analysis section would benefit from the inclusion of illustrative examples of failure cases, along with brief suggestions or hypotheses about how such challenges might be addressed in future work.

A brief discussion of potential ethical concerns or risks related to the deployment of large language models in educational contexts—such as bias, misinformation, or over-reliance—could add depth to the manuscript.

Lastly, while the data sources are clearly indicated, providing access to any code or implementation details (e.g., via a GitHub repository) would further support reproducibility and allow others to build upon this valuable work.

**Do you want your identity to be public for this peer review?** For information about this choice, including consent withdrawal, please see our Privacy Policy

Reviewer #1: No

Reviewer #2: **Yes: ** Muralidhar Kurni, Ph.D., PostDoc

---

## [Author Response · Author response to Decision Letter 1]

28 May 2025

Journal Requirements:

1.Please ensure that your manuscript meets PLOS ONE's style requirements, including those for file naming.

The PLOS ONE style templates can be found at

A: After comparing the standardised documents of the journals, this article conforms to the style and naming convention of the article.

A: The algorithms and simulation code covered in this paper are fully publicly available.

“This work is partially supported by 2025 Key Research Project of Shenzhen Polytechnic University "Research on Key Methods for Analysis and Prediction of Social Behaviour of Specific Characters on Multimodal Big Data (6025310008K)", 2024 Guangdong Province Education Science Planning Project (Higher Education Special Project) "Research on Smart Classroom Teaching Behavior Analysis Methods Based on Scene Semantic Understanding and Deep Learning Characteristic Representation (2024GXJK766)", 2023 Guangdong Provincial Higher Vocational Education Teaching Reform Research and Practice Project "Research on Online Teaching Quality Evaluation Method Based on Multimodal Affective State Analysis (2023JG277)", 2024 Shenzhen Polytechnic University Quality Engineering Project "Research on Classroom Scene Understanding and Behavior Analysis Method Based on Multimodal Attention Mechanisms (7024310268)", 2024 Higher Education Scientific Research Planning Project of the Chinese Society of Higher Education "Research on the Analysis of Teaching and Learning Deep Interaction Characteristics in Smart Classroom Environment Supported by Multimodal Data (24XH0407)", 2023 Shenzhen Education Science Planning Project"Research on the Evolutionary Mechanism and Intervention of Interpersonal Relationships among College Students Driven by Multimodal Data (rgzn23003)".

A: After careful combing, this paper mainly relies on the following six topics to carry out relevant research, and the algorithms proposed in the text are all supported by the following topics, including scientific research platforms, simulation data, and test applications.

1.2025 Guangdong Philosophy and Social Science Planning Project "Research on the Synergistic Evolutionary Mechanisms of the Governance System of Colleges and Universities and the Development of Students’ Socio-emotional Competence: Based on Multimodal Learning Analysis (GD25CJY29)".

2.2025 Key Research Project of Shenzhen Polytechnic University "Research on Key Methods for Analysis and Prediction of Social Behaviour of Specific Characters on Multimodal Big Data (6025310008K)", the research in this paper relies on this topic to carry out research aimed at solving the spatial mapping problem of multimodal heterogeneous features.

3.2024 Guangdong Province Education Science Planning Project (Higher Education Special Project) "Research on Smart Classroom Teaching Behavior Analysis Methods Based on Scene Semantic Understanding and Deep Learning Characteristic Representation (2024GXJK766)", based mainly on this topic, the research on behaviour detection is oriented towards the problem of joint mining for scene saliency regions and behaviour recognition.

4.2023 Guangdong Provincial Higher Vocational Education Teaching Reform Research and Practice Project "Research on Online Teaching Quality Evaluation Method Based on Multimodal Affective State Analysis (2023JG277)", based mainly on this topic, a behavioural prediction study for online teaching and learning is conducted by combining viewpoint summaries of teacher and student adaptive interaction behavioural events, propensity analysis and accurate modelling of online behavioural trajectories.

5.2024 Shenzhen Polytechnic University Quality Engineering Project "Research on Classroom Scene Understanding and Behavior Analysis Method Based on Multimodal Attention Mechanisms (7024310268)", in this study, a scene classification model based on the attention mechanism is proposed, theoretically validated and functionally tested in text.

6.2024 Higher Education Scientific Research Planning Project of the Chinese Society of Higher Education "Research on the Analysis of Teaching and Learning Deep Interaction Characteristics in Smart Classroom Environment Supported by Multimodal Data (24XH0407)", the definition and formal representation of the elements of teaching and learning activities in a smart classroom were investigated in that study based on automated quizzes, which were tested and validated in this paper.

“This work is partially supported by 2025 Key Research Project of Shenzhen Polytechnic University "Research on Key Methods for Analysis and Prediction of Social Behaviour of Specific Characters on Multimodal Big Data (6025310008K)", 2024 Guangdong Province Education Science Planning Project (Higher Education Special Project) "Research on Smart Classroom Teaching Behavior Analysis Methods Based on Scene Semantic Understanding and Deep Learning Characteristic Representation (2024GXJK766)", 2023 Guangdong Provincial Higher Vocational Education Teaching Reform Research and Practice Project "Research on Online Teaching Quality Evaluation Method Based on Multimodal Affective State Analysis (2023JG277)", 2024 Shenzhen Polytechnic University Quality Engineering Project "Research on Classroom Scene Understanding and Behavior Analysis Method Based on Multimodal Attention Mechanisms (7024310268)", 2024 Higher Education Scientific Research Planning Project of the Chinese Society of Higher Education "Research on the Analysis of Teaching and Learning Deep Interaction Characteristics in Smart Classroom Environment Supported by Multimodal Data (24XH0407)", 2023 Shenzhen Education Science Planning Project"Research on the Evolutionary Mechanism and Intervention of Interpersonal Relationships among College Students Driven by Multimodal Data (rgzn23003)".

A: Funding Disclaimer: No additional external funding was received for this study, and all research was conducted on the following six topics.

1.2025 Guangdong Philosophy and Social Science Planning Project "Research on the Synergistic Evolutionary Mechanisms of the Governance System of Colleges and Universities and the Development of Students’ Socio-emotional Competence: Based on Multimodal Learning Analysis (GD25CJY29)".

2.2025 Key Research Project of Shenzhen Polytechnic University "Research on Key Methods for Analysis and Prediction of Social Behaviour of Specific Characters on Multimodal Big Data (6025310008K)".

3.2024 Guangdong Province Education Science Planning Project (Higher Education Special Project) "Research on Smart Classroom Teaching Behavior Analysis Methods Based on Scene Semantic Understanding and Deep Learning Characteristic Representation (2024GXJK766)".

4.2023 Guangdong Provincial Higher Vocational Education Teaching Reform Research and Practice Project "Research on Online Teaching Quality Evaluation Method Based on Multimodal Affective State Analysis (2023JG277)".

5.2024 Shenzhen Polytechnic University Quality Engineering Project "Research on Classroom Scene Understanding and Behavior Analysis Method Based on Multimodal Attention Mechanisms (7024310268)".

6.2024 Higher Education Scientific Research Planning Project of the Chinese Society of Higher Education "Research on the Analysis of Teaching and Learning Deep Interaction Characteristics in Smart Classroom Environment Supported by Multimodal Data (24XH0407)".

5. We note that your Data Availability Statement is currently as follows: All relevant data are within the manuscript and its Supporting Information files.

A: The data and code applied in this paper uses only a portion of the data in the dataset, which can be accessed on request at https://github.com/hitlujin0808/education-Q-A

6. PLOS requires an ORCID iD for the corresponding author in Editorial Manager on papers submitted after December 6th, 2016. Please ensure that you have an ORCID iD and that it is validated in Editorial Manager. To do this, go to ‘Update my Information’ (in the upper left-hand corner of the main menu), and click on the Fetch/Validate link next to the ORCID field. This will take you to the ORCID site and allow you to create a new iD or authenticate a pre-existing iD in Editorial Manager.

A: The ORCIDs of the first and corresponding authors of this paper are registered, including the ORCID of JinLu (first author) is 0009-0004-5686-3654, and the ORCID of JiLi (corresponding author) is 0009-0003-3412-5493.

7. Please ensure that you refer to Figure 1, 3, 7, 8 and 9 in your text as, if accepted, production will need this reference to link the reader to the figure.

A: As revised, images 1, 3, 7, 8, and 9 are associated in the article with specific descriptions of their content.

8. Please upload a copy of Figure 4, to which you refer in your text on page 19 in PDF submission. If the figure is no longer to be included as part of the submission please remove all reference to it within the text.

A: As a result of the revision, figure 4 has been deleted and the presentation in the text has been adjusted and optimized accordingly.

9. We note you have included a table to which you do not refer in the text of your manuscript. Please ensure that you refer to Table 1, 2, 3, 5, 6 and 7 in your text; if accepted, production will need this reference to link the reader to the Table.

A: A total of seven tables are shown in this paper, and with some modifications, tables 1, 2, 3, 4, 5, 6, and 7 have been correlated in the text with corresponding descriptions.

Reviewers' comments:

Reviewer's Responses to Questions

Comments to the Author

1. Is the manuscript technically sound, and do the data support the conclusions?

Reviewer #1: Yes

Reviewer #2: Yes

2. Has the statistical analysis been performed appropriately and rigorously?

Reviewer #1: Yes

Reviewer #2: Yes

3. Have the authors made all data underlying the findings in their manuscript fully available?

Reviewer #1: Yes

Reviewer #2: Yes

4. Is the manuscript presented in an intelligible fashion and written in standard English?

Reviewer #1: Yes

Reviewer #2: Yes

5. Review Comments to the Author

Reviewer #1: In summary, this article is interesting and offers valuable findings in terms of both theoretical knowledge and practical application. However, there are several weaknesses that need to be addressed to enhance the overall quality of the article. Language errors are evident, and the writing style lacks consistency, particularly in the labelling of tables and figures. Some labels use lowercase letters while others capitalize each word, which disrupts uniformity. Additionally, the labels for Figures 1, 2, and 3 are unclear, and the colours used are not appropriate for academic writing.

A: Thank you for your valuable comments. Based on your suggestions, targeted changes have been made to the text as follows.

1.Grammatical issues in the text have been optimized, as detailed in red.

2.The formatting of all images and figure captions has been standardized in accordance with the journal's requirements.

3.Optimized description, title and colour scheme of images 1, 2, 3.

The font used throughout the article is also inconsistent across the main text, references, and figure labels. For instance, the citation for “Figure 4” in the text does not follow a consistent formatting style. Moreover, it is recommended that Figure 4 be converted into a tab

---

## [Decision Letter · Decision Letter 1]

10 Aug 2025

Dear Dr. Li,

Thank you for submitting your manuscript to PLOS One. After careful consideration, we feel that it has merit but does not fully meet PLOS One’s publication criteria as it currently stands. Therefore, we invite you to submit a revised version of the manuscript that addresses the points raised during the review process.

If applicable, we recommend that you deposit your laboratory protocols in protocols.io to enhance the reproducibility of your results. Protocols.io assigns your protocol its own identifier (DOI) so that it can be cited independently in the future. For instructions see: https://journals.plos.org/plosone/s/submission-guidelines#loc-laboratory-protocols . Additionally, PLOS One offers an option for publishing peer-reviewed Lab Protocol articles, which describe protocols hosted on protocols.io. Read more information on sharing protocols at https://plos.org/protocols?utm_medium=editorial-email&utm_source=authorletters&utm_campaign=protocols .

We look forward to receiving your revised manuscript.

Kind regards,

Hugh Cowley

Senior Editor

PLOS One

on behalf of

Vincenzo Bonnici, PhD

Academic Editor

PLOS One

Journal Requirements:

Reviewers' comments:

Reviewer's Responses to Questions

**Comments to the Author**

Reviewer #1: All comments have been addressed

Reviewer #2: All comments have been addressed

2. Is the manuscript technically sound, and do the data support the conclusions?

Reviewer #1: Yes

Reviewer #2: Yes

3. Has the statistical analysis been performed appropriately and rigorously?

Reviewer #1: Yes

Reviewer #2: (No Response)

4. Have the authors made all data underlying the findings in their manuscript fully available?

Reviewer #1: Yes

Reviewer #2: Yes

5. Is the manuscript presented in an intelligible fashion and written in standard English?

Reviewer #1: Yes

Reviewer #2: Yes

Reviewer #1: In general, the author has addressed the comments and made appropriate improvements accordingly. There is a noticeable improvement in the overall quality of the writing, reflecting the author’s effort to enhance clarity and coherence

Reviewer #2: The revised manuscript demonstrates significant improvements and adequately addresses all comments raised in the previous review round. The integration of Retrieval-Augmented Generation (RAG) and LLM-based Code Interpreters is well-motivated and technically justified. The experimental design is clear, and the four comparative configurations are now explicitly outlined early in the methodology section, enhancing reader comprehension.

The addition of the system architecture diagram (Figure 1) helps visualize the interaction between RAG, the Code Interpreter, and the LLM, which strengthens the conceptual clarity. The statistical significance tests introduced in Section 5.6.5 provide rigor to the performance claims, and the results in Table 8 confirm the robustness of the findings.

The authors have also improved reproducibility and transparency by sharing the implementation code via GitHub and including a detailed discussion of error handling within the sandboxed execution environment. The inclusion of a Chain-of-Thought example and illustrative failure cases further enriches the manuscript, offering practical insights into system behavior.

Finally, the newly added discussion on ethical concerns, covering bias, misinformation, and over-reliance, adds valuable depth and relevance, especially for educational applications of LLMs.

The manuscript is now clearly written, technically sound, and methodologically rigorous. I believe it is suitable for publication in its current form.

**Do you want your identity to be public for this peer review?** For information about this choice, including consent withdrawal, please see our Privacy Policy

Reviewer #1: No

Reviewer #2: **Yes: ** Muralidhar Kurni, Ph.D., PostDoc.

---

## [Author Response · Author response to Decision Letter 2]

24 Aug 2025

Journal Requirements:

A� We will carry out the paper revision work in strict accordance with the requirements.

A� Some references in the references have been replaced according to the requirements of the journal to enhance the accuracy of the citation. In addition, the DOI of the arXiv paper is supplemented to facilitate the paper query.

Reviewers' comments:

Reviewer's Responses to Questions

Comments to the Author

1. If the authors have adequately addressed your comments raised in a previous round of review and you feel that this manuscript is now acceptable for publication, you may indicate that here to bypass the “Comments to the Author” section, enter your conflict of interest statement in the “Confidential to Editor” section, and submit your "Accept" recommendation.

Reviewer #1: All comments have been addressed

Reviewer #2: All comments have been addressed

A� We would like to express our gratitude to the reviewers for their hard work.

2. Is the manuscript technically sound, and do the data support the conclusions?

Reviewer #1: Yes

Reviewer #2: Yes

A� We would like to express our gratitude to the reviewers for their hard work.

3. Has the statistical analysis been performed appropriately and rigorously?

Reviewer #1: Yes

Reviewer #2: (No Response)

A� We would like to express our gratitude to the reviewers for their hard work.

4. Have the authors made all data underlying the findings in their manuscript fully available?

Reviewer #1: Yes

Reviewer #2: Yes

A� We would like to express our gratitude to the reviewers for their hard work.

5. Is the manuscript presented in an intelligible fashion and written in standard English?

Reviewer #1: Yes

Reviewer #2: Yes

A� We would like to express our gratitude to the reviewers for their hard work.

6. Review Comments to the Author

Reviewer #1: In general, the author has addressed the comments and made appropriate improvements accordingly. There is a noticeable improvement in the overall quality of the writing, reflecting the author’s effort to enhance clarity and coherence

A Thank you again for your hard work, and we will continue to learn your valuable opinions and improve the quality of the paper.

Reviewer #2: The revised manuscript demonstrates significant improvements and adequately addresses all comments raised in the previous review round. The integration of Retrieval-Augmented Generation (RAG) and LLM-based Code Interpreters is well-motivated and technically justified. The experimental design is clear, and the four comparative configurations are now explicitly outlined early in the methodology section, enhancing reader comprehension.

A Thank you again for your hard work, and we will continue to learn your valuable opinions and improve the quality of the paper.

The addition of the system architecture diagram (Figure 1) helps visualize the interaction between RAG, the Code Interpreter, and the LLM, which strengthens the conceptual clarity. The statistical significance tests introduced in Section 5.6.5 provide rigor to the performance claims, and the results in Table 8 confirm the robustness of the findings.

A Thank you again for your hard work, and we will continue to learn your valuable opinions and improve the quality of the paper.

The authors have also improved reproducibility and transparency by sharing the implementation code via GitHub and including a detailed discussion of error handling within the sandboxed execution environment. The inclusion of a Chain-of-Thought example and illustrative failure cases further enriches the manuscript, offering practical insights into system behavior.

A Thank you again for your hard work, and we will continue to learn your valuable opinions and improve the quality of the paper.

Finally, the newly added discussion on ethical concerns, covering bias, misinformation, and over-reliance, adds valuable depth and relevance, especially for educational applications of LLMs.

A Thank you again for your hard work, and we will continue to learn your valuable opinions and improve the quality of the paper.

The manuscript is now clearly written, technically sound, and methodologically rigorous. I believe it is suitable for publication in its current form.

A Thank you again for your hard work, and we will continue to learn your valuable opinions and improve the quality of the paper.

7. PLOS authors have the option to publish the peer review history of their article (what does this mean?). If published, this will include your full peer review and any attached files.

Do you want your identity to be public for this peer review? For information about this choice, including consent withdrawal, please see our Privacy Policy.

Reviewer #1: No

Reviewer #2: Yes: Muralidhar Kurni, Ph.D., PostDoc.

A� We would like to express our gratitude to the reviewers for their hard work.

---

## [Decision Letter · Decision Letter 2]

22 Oct 2025

Dear Dr. Li,

Thank you for submitting your manuscript to PLOS ONE. After careful consideration, we feel that it has merit but does not fully meet PLOS ONE’s publication criteria as it currently stands. Therefore, we invite you to submit a revised version of the manuscript that addresses the points raised during the review process.

https://journals.plos.org/plosone/s/submission-guidelines#loc-laboratory-protocols . Additionally, PLOS ONE offers an option for publishing peer-reviewed Lab Protocol articles, which describe protocols hosted on protocols.io. Read more information on sharing protocols at https://plos.org/protocols?utm_medium=editorial-email&utm_source=authorletters&utm_campaign=protocols .

We look forward to receiving your revised manuscript.

Kind regards,

Zheng Zhang

Academic Editor

PLOS ONE

Journal Requirements:

Comments from the Editorial Office:

In the previous decision letter, the PLOS One Editorial Office requested that you address the following points, which remain unaddressed. Please give further attention to these requirements before submitting a revised manuscript and response document:

1) We thank you for providing a link to the repository from which the code underpinning this study can be freely accessed (https://github.com/hitlujin0808/education-Q-A). We note that this link has been provided in your Response to Reviewers file, but does not appear to have been included in your manuscript text. Please revise your manuscript text to include this link.

2) We note that the documentation for the code in the repository you have linked is provided in a language other than English. Because PLOS One is an international journal published in English, we kindly ask that you update the documentation in this repository to include translation into English. This is to ensure that your study complies with the PLOS One policy on code sharing, which states that you are encouraged to share your code 'in a way that follows best practice and facilitates reproducibility and reuse' (https://journals.plos.org/plosone/s/materials-software-and-code-sharing#loc-sharing-code).

Reviewers' comments:

Reviewer's Responses to Questions

**Comments to the Author**

Reviewer #1: All comments have been addressed

Reviewer #2: All comments have been addressed

2. Is the manuscript technically sound, and do the data support the conclusions?

Reviewer #1: Yes

Reviewer #2: Yes

3. Has the statistical analysis been performed appropriately and rigorously?

Reviewer #1: Yes

Reviewer #2: Yes

4. Have the authors made all data underlying the findings in their manuscript fully available?

Reviewer #1: (No Response)

Reviewer #2: Yes

5. Is the manuscript presented in an intelligible fashion and written in standard English?

Reviewer #1: (No Response)

Reviewer #2: Yes

Reviewer #1: COMMENTS.

SO FAR ONLY THOSE COMMENTS HAVE SYSTEM ERROR!

PAGE 11_ 1. Introduction

The educational question-answering systemError! Reference source not found. is playing an increasingly important role in modern education.

PAGE 12_ However, despite their excellence in text generation and comprehension, LLMs still face limitations in handling complex reasoning and mathematical calculations, and they are susceptible to the “hallucination[8]” problem, where the model may generate incorrect answers that do not align with actual knowledge, posing challenges to the accuracy and reliability of the educational systemError! Reference source not found..

To address these challenges in current educational question-answering systems, this study proposes a new LLM-based educational question-answering system that combines RetrievalAugmented Generation[9][11]Error! Reference source not found. (RAG) and LLM Code Interpreter[12][14][15][16].

PAGES 14_ 2 Related works

2.1 Knowledge Base Question Answering Systems (KBQA)

Past educational question answering systems primarily relied on Knowledge Base Question Answering Systems[20][21]Error! Reference source not found.

PAGES15_ Second, methods based on structured knowledge bases still face considerable challenges in dealing with the diversity and complexity of natural languageError! Reference source not found..

PAGES 16_ it may still generate inaccurate or erroneous results when confronted with tasks that require multi-step calculations or precise answersError! Reference source not found..

PAGES 16_2.3 Large Language Models (LLMs) and Code Interpreters

To address the limitations of LLMs in reasoning and computation, researchers have gradually introduced the Code Interpreter component[16][32]. A Code Interpreter is a tool that can generate and execute code, allowing it to handle complex tasks such as mathematical operations[34], data manipulation[35], and logical reasoningError! Reference source not found.

PAGES 18_ 4. Textbook Question Answering (TQA)

The TQAError! Reference source not found. dataset comes from high school science textbooks and includes 1,076 course contents from life sciences, Earth sciences, and physical sciences.

PAGES 21_ 4.4 Implementation Details of the RAG Retrieval Module

The RAG mechanism was implemented using the following approach, including knowledge base construction, retrieval and text selection, as well as fusion and input. The knowledge base was constructed by integrating publicly available text sources, such as Wikipedia excerpts, popular science websites, and digitized versions of various textbooks in both Chinese and English to ensure

linguistic consistency in retrieval. The original text was segmented into sentences or paragraphs, and vector-based retrieval techniquesError! Reference source not found. (e.g., Faiss or Milvus) were used to index the text

PAGES 22_ 4.6 Prompt Engineering

To maximize the synergistic effects of the LLM, RAG, and Code Interpreter components, careful prompt engineering techniques[44][45]Error! Reference source not found. were employed.

Reviewer #2: The manuscript has undergone substantial improvement in both clarity and methodological rigor compared to earlier versions.

The integration of Retrieval-Augmented Generation (RAG) and LLM-based Code Interpreters is well-motivated, technically justified, and convincingly demonstrated across diverse educational datasets.

The addition of the system architecture diagram, statistical significance testing, and error analysis has strengthened the scientific quality and credibility of the work.

Sharing the implementation code via GitHub enhances transparency, reproducibility, and the practical applicability of the study.

The discussion of ethical considerations—bias, misinformation, and over-reliance—is highly relevant and adds value to the paper, especially in educational contexts.

The writing is clear, coherent, and presented in standard English, making the manuscript accessible to a broad readership.

Overall, the research is technically sound, the data support the conclusions, and the paper is suitable for publication in its current form.

**Do you want your identity to be public for this peer review?**  For information about this choice, including consent withdrawal, please see our Privacy Policy

Reviewer #1: No

Reviewer #2: **Yes: ** Muralidhar Kurni

---

## [Author Response · Author response to Decision Letter 3]

26 Oct 2025

Journal Requirements:

A Thank you for your suggestions. I shall carefully review the reviewers' comments regarding the references and promptly make the necessary improvements.

A Thank you for your suggestion. All references have been checked and are indeed articles from the database.

Comments from the Editorial Office:

In the previous decision letter, the PLOS One Editorial Office requested that you address the following points, which remain unaddressed. Please give further attention to these requirements before submitting a revised manuscript and response document:

1) We thank you for providing a link to the repository from which the code underpinning this study can be freely accessed (https://github.com/hitlujin0808/education-Q-A). We note that this link has been provided in your Response to Reviewers file, but does not appear to have been included in your manuscript text. Please revise your manuscript text to include this link.

A Thank you for your suggestion. The code link has been added at the end of the paper.

2) We note that the documentation for the code in the repository you have linked is provided in a language other than English. Because PLOS One is an international journal published in English, we kindly ask that you update the documentation in this repository to include translation into English. This is to ensure that your study complies with the PLOS One policy on code sharing, which states that you are encouraged to share your code 'in a way that follows best practice and facilitates reproducibility and reuse' (https://journals.plos.org/plosone/s/materials-software-and-code-sharing#loc-sharing-code).

A Thank you for your suggestion. The explanatory notes for the code have now been fully converted into English.

Reviewers' comments:

Reviewer's Responses to Questions

Comments to the Author

1. If the authors have adequately addressed your comments raised in a previous round of review and you feel that this manuscript is now acceptable for publication, you may indicate that here to bypass the “Comments to the Author” section, enter your conflict of interest statement in the “Confidential to Editor” section, and submit your "Accept" recommendation.

Reviewer #1: All comments have been addressed

A Thank you for your support.

Reviewer #2: All comments have been addressed

A Thank you for your support.

2. Is the manuscript technically sound, and do the data support the conclusions?

Reviewer #1: Yes

A Thank you for your support.

Reviewer #2: Yes

A Thank you for your support.

3. Has the statistical analysis been performed appropriately and rigorously?

Reviewer #1: Yes

A Thank you for your support.

Reviewer #2: Yes

A Thank you for your support.

4. Have the authors made all data underlying the findings in their manuscript fully available?

Reviewer #1: (No Response)

A Thank you for your support.

Reviewer #2: Yes

A Thank you for your support.

5. Is the manuscript presented in an intelligible fashion and written in standard English?

Reviewer #1: (No Response)

A Thank you for your support.

Reviewer #2: Yes

A Thank you for your support.

6. Review Comments to the Author

Reviewer #1: COMMENTS.

SO FAR ONLY THOSE COMMENTS HAVE SYSTEM ERROR!

PAGE 11_ 1. Introduction

The educational question-answering systemError! Reference source not found. is playing an increasingly important role in modern education.

A Thank you for your suggestion. The issue has been resolved by replacing the reference 1&2.

PAGE 12_ However, despite their excellence in text generation and comprehension, LLMs still face limitations in handling complex reasoning and mathematical calculations, and they are susceptible to the “hallucination[8]” problem, where the model may generate incorrect answers that do not align with actual knowledge, posing challenges to the accuracy and reliability of the educational systemError! Reference source not found..

A Thank you for your suggestion. The issue has been resolved by replacing the reference 8.

To address these challenges in current educational question-answering systems, this study proposes a new LLM-based educational question-answering system that combines RetrievalAugmented Generation[9][11]Error! Reference source not found. (RAG) and LLM Code Interpreter[12][14][15][16].

A Thank you for your suggestion. The issue has been resolved by replacing the reference 10&11.

PAGES 14_ 2 Related works

2.1 Knowledge Base Question Answering Systems (KBQA)

Past educational question answering systems primarily relied on Knowledge Base Question Answering Systems[20][21]Error! Reference source not found.

A Thank you for your suggestion. The issue has been resolved by replacing the reference 20&21.

PAGES15_ Second, methods based on structured knowledge bases still face considerable challenges in dealing with the diversity and complexity of natural languageError! Reference source not found..

A Thank you for your suggestion. The issue has been resolved by replacing the reference 22.

PAGES 16_ it may still generate inaccurate or erroneous results when confronted with tasks that require multi-step calculations or precise answersError! Reference source not found..

A Thank you for your suggestion. The issue has been resolved by replacing the reference 32.

PAGES 16_2.3 Large Language Models (LLMs) and Code Interpreters

To address the limitations of LLMs in reasoning and computation, researchers have gradually introduced the Code Interpreter component[16][32]. A Code Interpreter is a tool that can generate and execute code, allowing it to handle complex tasks such as mathematical operations[34], data manipulation[35], and logical reasoningError! Reference source not found.

A Thank you for your suggestion. The issue has been resolved by replacing the reference 36.

PAGES 18_ 4. Textbook Question Answering (TQA)

The TQAError! Reference source not found. dataset comes from high school science textbooks and includes 1,076 course contents from life sciences, Earth sciences, and physical sciences.

A Thank you for your suggestion. The issue has been resolved by replacing the reference 41.

PAGES 21_ 4.4 Implementation Details of the RAG Retrieval Module

The RAG mechanism was implemented using the following approach, including knowledge base construction, retrieval and text selection, as well as fusion and input. The knowledge base was constructed by integrating publicly available text sources, such as Wikipedia excerpts, popular science websites, and digitized versions of various textbooks in both Chinese and English to ensure

linguistic consistency in retrieval. The original text was segmented into sentences or paragraphs, and vector-based retrieval techniquesError! Reference source not found. (e.g., Faiss or Milvus) were used to index the text

A Thank you for your suggestion. The issue has been resolved by replacing the reference 46.

PAGES 22_ 4.6 Prompt Engineering

To maximize the synergistic effects of the LLM, RAG, and Code Interpreter components, careful prompt engineering techniques[44][45]Error! Reference source not found. were employed.

A Thank you for your suggestion. The issue has been resolved by replacing the reference 45&46&49.

Reviewer #2: The manuscript has undergone substantial improvement in both clarity and methodological rigor compared to earlier versions.

A Thank you for your support.

The integration of Retrieval-Augmented Generation (RAG) and LLM-based Code Interpreters is well-motivated, technically justified, and convincingly demonstrated across diverse educational datasets.

A Thank you for your support.

The addition of the system architecture diagram, statistical significance testing, and error analysis has strengthened the scientific quality and credibility of the work.

A Thank you for your support.

Sharing the implementation code via GitHub enhances transparency, reproducibility, and the practical applicability of the study.

A Thank you for your support.

The discussion of ethical considerations—bias, misinformation, and over-reliance—is highly relevant and adds value to the paper, especially in educational contexts.

A Thank you for your support.

The writing is clear, coherent, and presented in standard English, making the manuscript accessible to a broad readership.

A Thank you for your support.

Overall, the research is technically sound, the data support the conclusions, and the paper is suitable for publication in its current form.

A Thank you for your support.

7. PLOS authors have the option to publish the peer review history of their article (what does this mean?). If published, this will include your full peer review and any attached files.

Do you want your identity to be public for this peer review? For information about this choice, including consent withdrawal, please see our Privacy Policy.

Reviewer #1: No

A Thank you for your support.

Reviewer #2: Yes: Muralidhar Kurni

A Thank you for your support.

---

## [Editor Report · Decision Letter 3]

7 Nov 2025

A Novel Framework for Educational Q&A: Leveraging RAG and Code Interpreters for Knowledge Retrieval and Logical Computation

PONE-D-25-10858R3

Dear Dr. Ji Li,

We’re pleased to inform you that your manuscript has been judged scientifically suitable for publication and will be formally accepted for publication once it meets all outstanding technical requirements.

Kind regards,

Zheng Zhang

Academic Editor

PLOS ONE

Additional Editor Comments (optional):

I believe the revisions have been made thoroughly, and the manuscript can be accepted.
---

## [Editor Report · Acceptance letter]

PONE-D-25-10858R3

PLOS ONE

Dear Dr. Li,

I'm pleased to inform you that your manuscript has been deemed suitable for publication in PLOS ONE. Congratulations! Your manuscript is now being handed over to our production team.

Kind regards,

on behalf of

Dr. Zheng Zhang

Academic Editor

PLOS ONE